# ADAPTIVE DEPTH TSETLIN AUTOMATON

## ABSTRACT

The Tsetlin Automaton (TA) is a foundational single-state reinforcement learning model, but its fixed depth parameter ($N$) poses a significant limitation for navigating the exploration and exploitation dilemma. Despite remarkable advancements, existing TA models lack adaptability in real-world scenarios where dynamic depth adjustments are essential. In this paper, we introduce the Adaptive Depth Tsetlin Automaton (ADTA), a novel solution addressing this challenge. ADTA integrates TA with a reinforcement agent capable of dynamically modifying $N$. We analyze ADTA using Lyapunov stability theorem and Markov chain analysis within a dual-environment framework: the outer environment, where TA operates to maximize rewards, and the inner environment, where a reinforcement learning agent evaluates TA's performance based on $N$. Through actions like 'Grow,' 'Shrink,' and 'Stop,' the inner agent configures $N$ dynamically. Unlike conventional TA configurations with fixed $N$, our approach demonstrates improved reward maximization and regret minimization. Furthermore, we present numerical simulations that corroborate our theoretical results.

## 1 INTRODUCTION

The *Tsetlin Automaton (TA)* (Tsetlin, 1961; Narendra & Thathachar, 2012) is an innovative concept in a single-state reinforcement learning (Sutton & Barto, 2018; Çalışır & Pehlivanoğlu, 2019; Zhang et al., 2022), designed to capture the intricate nature of human decision-making and calculation Narendra & Thathachar (2012). Inspired by principles from psychology, it emulates human-like learning strategies within computational frameworks. Notably, the TA represents a pioneering solution to the well-known multi-armed bandit problem (Yuan et al., 2022; Amani & Thrampoulidis, 2021; Ramponi et al., 2021) and serves as the first learning algorithm in *the Learning Automaton (LA)* (Granmo, 2018; Abeyrathna et al., 2020; Belaid et al., 2023) family.

The TA (Narendra & Thathachar, 2012; Granmo, 2018) is characterized by its state-machine architecture, featuring a grid-like configuration of $K$ actions and a fixed depth parameter $N$ for each action (number of nodes in each action), illustrated in Figure 1. Within the TA, each node corresponds to a specific action and is identified by a unique pair. This design allows the automaton to transition between nodes based on rewards from its environment seamlessly. When receiving a reward, the chosen action is reinforced by transitioning from higher-numbered nodes to lower-numbered nodes within the action's depth, thereby increasing its likelihood of selection. Conversely, in case of a penalty, the transition occurs from lower-numbered nodes to higher-numbered ones within the action. If the highest-numbered node is reached and a penalty is incurred, the next action is chosen according to a clockwise policy for the subsequent iteration. For example, $\phi_{(3,2)}$ denotes the first node of action 3. Upon receiving a reward, the TA transitions to $\phi_{(3,1)}$ to increase the likelihood of selecting action 3 twice. Conversely, a penalty leads the TA to transition to $\phi_{(4,2)}$, indicating that selecting action 3 was incorrect and its selection chances are exhausted. This structured framework enables the TA to effectively capture and represent the nuances of the learning process within a fixed architecture.

TAs are typically employed in online learning decision-making scenarios, making it impractical to adjust their depth using offline data (Narendra & Thathachar, 2012; Granmo, 2018). Therefore, an online mechanism is essential for adapting the TA's depth based on specific problem requirements, allowing for incremental, decremental, or unchanged adjustments. This mechanism will help the TA to explore and exploit the environment efficiently. Lower depth values may result in more exploration and frequent action switching. Conversely, higher depth values may lead to challenges in

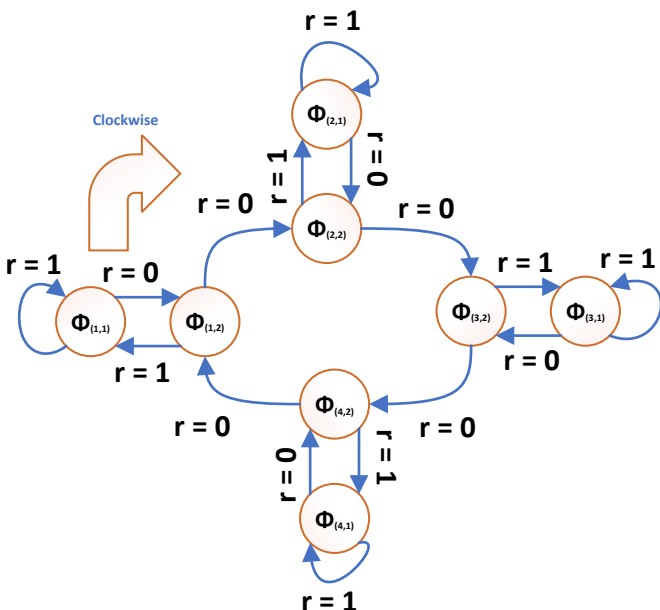

Figure 1: A four-action Tsetlin Automaton with depth two for each action

learning, as it becomes difficult to modify actions effectively, more exploitation toward sub-optimal action and diminishing the TA's efficacy. Given this critical challenge in the current implementation, it is essential to adaptively adjust the TA's depth to optimize performance (Narendra & Thathachar, 2012).

Moreover, from the perspective of Automated Machine Learning (AutoML) (He et al., 2021; Feurer et al., 2015; Kübler et al., 2022), the TA lacks internal mechanisms or tools to discern whether the selected depth is suitable or requires adjustment based on environmental conditions (Narendra & Thathachar, 2012; Granmo, 2018; Abeyrathna et al., 2020). Hence, there is a need for another learning agent to be integrated with the TA. This auxiliary agent can assist the TA in modifying its depth based on rewards and penalties received from the environment.

In this paper, we introduce the *Adaptive Depth Tsetlin Automaton (ADTA)*, a novel approach that intelligently learns the appropriate depth for the TA. By employing a central reinforcement learning agent, the ADTA dynamically adjusts the depth of all actions within the automaton. This self-adaptive, flexible methodology enables the TA to strike an ideal balance between exploration and exploitation, significantly improving its performance and adaptability in a wide range of complex environments.

## 1.1 CONTRIBUTIONS

- We identify a critical parameter $N$, which represents the depth parameter in the TA. This parameter poses a challenge in balancing the exploration and exploitation capabilities of the TA.

- We propose a learning algorithm aimed at dynamically adjusting the depth parameter within the TA. This algorithm is devised by integrating the TA with an RL agent. During each epoch, the algorithm assesses the current value of $N$, and when the TA seeks to alter its action, the RL agent intervenes to adjust the depth accordingly.

- We offer a theoretical analysis that demonstrates the learning ability of the ADTA using a combination of Markov process and Lyapunov stability theorem.

- We complement our theoretical results with numerical simulations and corresponding discussions on the performance of our algorithm.

- We illustrate ADTA's practical versatility by applying it to the dropout problem in deep neural networks, effectively countering overfitting concerns.

- We demonstrate the effectiveness of our learning model in a highly relevant and practical context by applying ADTA to Bitcoin, a leading decentralized cryptocurrency. ADTA is deployed as a distributed, real-time decision-making mechanism to mitigate the impact of selfish mining attacks.

## 1.2 PROBLEM FORMULATION

**TA Model.** The TA consists of $K$ actions denoted as $a_1, a_2, \ldots, a_K$, and $KN$ nodes represented by $\phi_{(1,1)}, \phi_{(1,2)}, \ldots, \phi_{(1,N)}, \ldots, \phi_{(K,N)}$. Each node is defined by an ordered pair $(i, j)$, where $1 \leq i \leq K$ indicates the action number, and $1 \leq j \leq N$ denotes the node number. When the TA is in node $\phi_{(i,j)}$, it performs action $A(n) = a_i$ in the $n^{th}$ iteration. The environment will respond to the TA with $r$, where $r \in R = \{0, 1\}$. In the event of an unfavorable response (i.e., $R(n) = r = 0$), the state transitions occur as follows:

$$\begin{cases} \phi_{(i,j)} \to \phi_{(i,j+1)} & (1 \leq j \leq N-1) \\ \phi_{(i,j)} \to \phi_{(i+1,j)} & (j = N) \end{cases} \tag{1}$$

Similarly, in the case of a favorable response (i.e., $R(n) = r = 1$), the state transitions are determined by:

$$\begin{cases} \phi_{(i,j)} \to \phi_{(i,j-1)} & (2 \leq j \leq N) \\ \phi_{(i,1)} \to \phi_{(i,1)} & (j = 1) \end{cases} \tag{2}$$

The selection of the next action in this automaton follows a clockwise pattern ($\phi_{(i,j)} \to \phi_{(i+1,j)}$).

**Goal.** In scenarios where the TA requires a change in action due to receiving a penalty from the environment ($\phi_{(i,j)} \to \phi_{(i+1,j)}$), the current depth parameter $N$ may not be appropriately configured. Thus, the main objective becomes either to increase ($N \leftarrow N + 1$) for exploitation, decrease ($N \leftarrow N - 1$) for exploration, or maintain the depth unchanged in such situations.

## 1.3 RELATED WORKS

**Tsetlin Automaton**. Drawing inspiration from Sutton and Barto's influential book (Sutton & Barto, 2018) on reinforcement learning, the exploration of learning automaton (LA) has played a pivotal role in shaping modern research, particularly in the domain of *trial-and-error* learning. Among the variants of LA, the TA (Granmo, 2018; Granmo et al., 2019) stands out as a prominent model that offers valuable insights into human decision-making processes and cognitive mechanisms. Notably, its recent integration with Neural Networks (Sharma et al., 2023; Seraj et al., 2022; Abeyrathna et al., 2021b; Bhattarai et al., 2022; Darshana Abeyrathna et al., 2020; Phoulady et al., 2019; Abeyrathna et al., 2021a; 2020; Glimsdal & Granmo, 2021; Bhattarai et al., 2023) has gained remarkable attention from researchers. Furthermore, over its renowned half-century existence, it has demonstrated its versatility and applicability across a wide spectrum of fields (Narendra & Thathachar, 2012) including: decentralized control (Tung & Kleinrock, 1996), searching on the line (Oommen, 1997), equi-partitioning (Oommen & Ma, 1988), streaming sampling for social activity networks (Ghavipour & Meybodi, 2018), faulty dichotomous search (Yazidi & Oommen, 2018), learning in deceptive environments (Zhang et al., 2016), and routing in telecommunication networks (Oommen et al., 2007).

Despite the TA's effectiveness in uncertain and stochastic environments where problems require quick trial-and-error solutions, it faces two fundamental challenges affecting its performance across various applications (Narendra & Thathachar, 2012): (1) The clockwise policy for selecting the next action; (2) The fixed depth value.

**Hybrid Tsetlin Automaton**. Gholami et al. (Gholami et al., 2023) proposed a solution to replace the clockwise policy of choosing the next action in TA by integrating it with an RL agent. This method is called the Hybrid Learning Automaton or HLA. In this method, an RL agent will learn

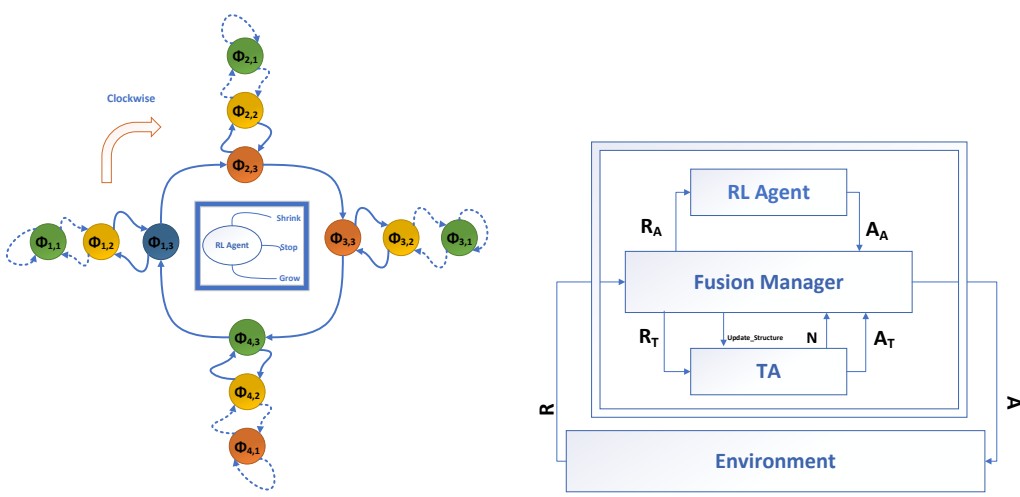

(a) ADTA with the RL agent modifying TA's structure: The RL agent's action can increase depth (Green), maintain current depth (Yellow), or decrease depth (Red) of TA.

(b) Architecture of ADTA

Figure 2: An overview of ADTA

how to choose the next action in order to maximize the cumulative reward of HLA without changing the structure of the TA, especially $N$. Despite the proposed model does not solve the fixed depth problem, it can be considered as the *state-of-the-art* method in the TA field.

For a deeper exploration of the LA family and its position within the realm of model-free reinforcement learning, please refer to Sections Appendix B and Appendix C.

## 2 ADAPTIVE DEPTH TSETLIN AUTOMATON

In this section, we introduce two key quantities: (1) $N$, the depth of the ADTA, representing the number of nodes in each action of the ADTA; (2) *Transition*, denoted by $\rightarrow$, illustrating how the ADTA moves from one node to another. There are four types of transitions: (i) $\phi_{(i,j)} \rightarrow \phi_{(i,j-1)}$ indicates a traverse to deeper nodes due to a reward; (ii) $\phi_{(i,j)} \rightarrow \phi_{(i,j+1)}$ indicates a traverse from deeper nodes to outer nodes due to a penalty; (iii) $\phi_{(i,N)} \rightarrow \phi_{(i+1,N)}$ indicates a traverse from one action to the next; (iv) $\phi_{(i,1)} \rightarrow \phi_{(i,1)}$ or $\phi_{(i,2)} \rightarrow \phi_{(i,1)}$ indicates depth traversal. For instance, in Figure 2a, the TA of ADTA has three actions with a depth ($N$) of three. Considering the transitions in (3): transition (1) is of type (i); transitions (2) and (3) are of type (iv); transitions (4) and (5) are of type (ii); and transition (6) is of type (iii).

$$\phi_{(3,3)} \xrightarrow{(1)} \phi_{(3,2)} \xrightarrow{(2)} \phi_{(3,1)} \xrightarrow{(3)} \phi_{(3,1)} \xrightarrow{(4)} \phi_{(3,2)} \xrightarrow{(5)} \phi_{(3,3)} \xrightarrow{(6)} \phi_{(4,3)} \tag{3}$$

Based on these considerations, we propose an adaptive TA algorithm called *ADTA*. The ADTA comprises three units (Figure 2b): (i) TA, serving as the foundation of ADTA; (ii) an RL agent tasked with controlling the depth of the TA; and (iii) Fusion Manager, which acts as the coordinator between the TA and the RL agent to set an appropriate depth.

**TA**. This unit operates similarly to the *Tsetlin Automaton*, as described in detail in Section 1.2. It selects action $A_T$ based on the current node ($\phi_{(i,j)}$) and communicates this choice to the Fusion Manager unit. In response to its selected action, it receives $R_T \in \{0, 1\}$.

**RL Agent**. This unit comprises an RL agent with three actions: $'Grow', 'Stop', 'Shrink'$. Choosing the *'Grow'* action leads to $N \leftarrow N + 1$, while selecting *'Stop'* maintains $N$ constant, and opting for *'Shrink'* yields to $N \leftarrow N - 1$. The Fusion Manager unit evaluates the selected depth and provides

---

**Algorithm 1** ADTA Learning Algorithm

---

**Notation:** TA with K actions and N nodes, the RL Agent
1: **Begin**
2:    **for** all episodes **do**
3:        Play $A_T \in \{1, ..., K\}$ of the TA through $A \in \{1, ..., K\}$
4:        Observe $R \in \{0, 1\}$ from the environment
5:        Compute $R_T \in \{0, 1\}$ for TA
6:        Observe Action Traverse ($\phi_{(i,N)} \rightarrow \phi_{(i+1,N)}$)
7:        **if** Action Traverse **then**
8:            Activate the RL agent
9:            Compute $R_A \in [0, 1]$ using (4)
10:           Observe the last value of $N$
11:           Play $A_A \in \{'Grow', 'Stop', 'Shrink'\}$
12:           Play $Update\_Structure$ with new N
13:       **end if**
14:   **end for**
15: **End**

---

reward or penalty feedback via $R_A \in [0, 1]$ using (4).

$$R_{RL\,Agent} = \frac{Number\ of\ Depth\ Transitions\ in\ i^{th}\ Action}{Total\ Number\ of\ Transitions\ in\ i^{th}\ Action} \tag{4}$$

**Fusion Manager**. This unit coordinates the collaboration between the TA and the RL agent. In each epoch, it receives $A_T$ from the TA unit, which it then designates as the final action of the ADTA ($A$ in Figure 2b). The environment responds to this action in the form of $R \in \{0, 1\}$. Initially, the Fusion Manager relays this response to the TA unit, prompting the TA to transition to a new node. If this transition changes the selected action, $N$ is transmitted to the Fusion Manager, which computes $R_A$. Subsequently, it awaits the RL agent unit to receive the updated $N$ value, triggering a structural adjustment in the TA (*Update_Structure* in Figure 2b).

With the above definitions and explanations, we are now ready to summarize ADTA in Algorithm 1. Moreover, to provide a visual representation, Figure 2a illustrates a ADTA with initially four actions and two nodes per action (Without considering dotted nodes at first). This configuration corresponds to TA with $K = 4$ and $N = 2$, where action 1 is chosen (highlighted in blue). In the event of an unfavorable response, the ADTA needs to rotate clockwise and select action 2. However, prior to that, the RL agent determines the appropriate depth for ADTA. If the RL agent opts for 'Grow', the number of states per action will increase by one, transforming ADTA with $N = 2$ into ADTA with $N = 3$ (indicated by the green state in Figure 2a). Conversely, selecting 'Shrink' by the RL agent will change ADTA with $N = 2$ to ADTA with $N = 1$ (represented by the red state in Figure 2a). Lastly, choosing 'Stop' will maintain the existing structure (depicted by the yellow state in Figure 2a). For clearer illustration, numerical examples are provided in Section D.

In the subsequent sections, we conduct a theoretical analysis of our proposed learning algorithm. In Section 2.1, we partition the ADTA's environment into internal, relating to the RL agent, and external, concerning the TA. Section 2.2 outlines the necessary definitions. We explore the theoretical foundation for determining the convergence point of the RL agent and analyzing the internal environment in Section 2.3. Lastly, we prove the learning capability of ADTA through theoretical analysis of the external environment in Section 2.4.

## 2.1 ENVIRONMENT SEPARATION

The ADTA integrates two distinct RL agents: the TA and an RL agent with three actions. To assess its learning capacity, we conduct a theoretical analysis in two discrete environments, termed as *internal* and *external* (as illustrated in Figure 3). By going deeper into the internal environment, we aim to identify the RL agent's convergence point using Lyapunov stability theorem. This insight enables the evaluation of TA's learning capability after determining the appropriate depth through Markov analysis. It's worth noting that we assume ADTA has two actions and an initial depth of $N$.

## 2.2 REQUIRED DEFINITIONS

This section has been dedicated to introducing some prerequisite definitions of the proof. All of these definitions are related to the concept of learning ability in the Tsetlin Automaton.

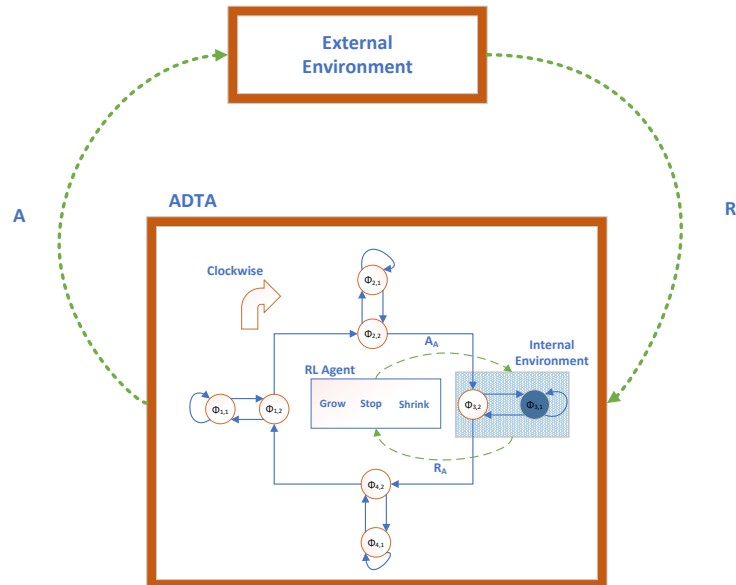

Figure 3: Separation of external and internal environment from each other in ADTA

**Definition 1.** *$M(n)$ is defined as the average penalty of a Tsetlin Automaton with $K$ actions and action probability $p(n)$ in a stationary environment. The environment provides responses of $R(n) \in \{0, 1\}$ in the $n^{th}$ iteration, where a response of 0 indicates a penalty and 1 indicates a reward. The penalty probabilities are constant and denoted as $\{c_1, c_2, ..., c_K\}$. The $M(n)$ is calculated as follows:*

$$M(n) = E[R(n)|p(n)] \tag{5}$$

$$= \sum_{i=1}^{K} \underbrace{P_r[R(n) = 0|A(n) = a_i]}_{Constant\ value\ equals\ to\ c_i} \underbrace{P_r[A(n) = a_i]}_{Probability\ of\ each\ action}$$

$$M(n) = \sum_{i=1}^{K} c_i p_i(n) \tag{6}$$

**Definition 2.** *The Pure Chance Automaton (PCA) is an automaton in which each action is chosen with equal probability. The action probability vector $p(n)$ for a PCA with $K$ actions is defined as follows:*

$$p_i(n) = \frac{1}{K} \qquad i \in \{1, 2, \cdots, K\} \tag{7}$$

**Definition 3.** *For a pure chance automaton with $K$ actions, $M(n)$ becomes a constant parameter denoted as $M_0$, defined by:*

$$M_0 = \frac{1}{K} \sum_{i=1}^{K} c_i \tag{8}$$

*This value equals $\frac{1}{2}(c_1 + c_2)$ for a pure chance automaton with two actions.*

**Definition 4.** *A Tsetlin Automaton is capable to learn if*

$$\lim_{n \to \infty} E[M(n)] < M_0 \tag{9}$$

### 2.3 INTERNAL ENVIRONMENT ANALYSIS

Firstly, we start our analysis by introducing the concept of the internal environment in which the RL agent interacts with the TA. Then, we aim to find the convergence point of the RL agent at which the appropriate depth for TA is chosen.

**Definition 5.** *The internal environment is defined as the environment in which the RL agent interacts with the TA by switching from the current action to a new action. This process continues as the agent makes decisions about the new depth of the TA. The internal environment of the RL agent with three actions is defined as follows:*

$$C_{int} = \{c_{int-Grow}, c_{int-Stop}, c_{int-Shrink}\} \tag{10}$$

*Additionally, each item (for instance $j^{th}$ item) in $C_{int}$ is a function:*

$$c_{int-j}(n) = f(n, C_{ext}, T, N) \tag{11}$$

*In the above function, $n$ denotes the iteration number, $C_{ext}$ denotes an external environment, $T$ denotes the number of transitions (depth and total transitions in the $i^{th}$ action), and $N$ denotes the depth of the Tsetlin Automaton in the current iteration.*

**Assumption 1.** *Let $\Delta$ be a threshold according to the properties of the internal environment in Definition 5. Then, for all $n \geq \Delta$, the internal environment turns into an environment with negligible changes in the penalty probability. These changes can be expressed as follows:*

$$\exists \Delta, \quad |c_{int-j}(n+1) - c_{int-j}(n)| = 0 \quad \forall n \geq \Delta \tag{12}$$

**Lemma 1.** *Let $\Delta$ exists for the internal environment, then for $n \geq \Delta$, the internal environment can be a stationary environment in which the penalty set will be defined as follows:*

$$C_{int} = \{\psi_{int-Grow}, \psi_{int-Stop}, \psi_{int-Shrink}\} \tag{13}$$

*Proof.* According to the assumption 1, since $n \geq \Delta$, thus $n$ exceeds the defined boundary criteria. This leads to the assumption that the internal environment has turned into a stationary environment. Therefore, we can consider the penalty probability of each RL agent's action equal to the $\psi_j$ constant value. As we know before from definition 5, the penalty set is compromised of three values that will be replaced by $\psi_j$ constants named $\psi_{int-Grow}$, $\psi_{int-Stop}$, and $\psi_{int-Shrink}$.

$\square$

Since it is proved that the internal environment has turned to the stationary environment, we can claim that the impact of the internal environment on the RL agent is independent of the iteration number. Therefore, the RL agent can be analyzed as a Markov process. Also, we will assume that the internal environment has an absorbing state.

**Definition 6.** *The state space of Markov process for the RL agent is defined to be:*

$$\Omega = \{p \mid [p_{Grow}, p_{Stop}, p_{Shrink}], 0 \leq p_{Grow} \leq 1, 0 \leq p_{Stop} \leq 1,$$
$$0 \leq p_{Shrink} \leq 1, p_{Grow} + p_{Stop} + p_{Shrink} = 1\} \tag{14}$$

**Definition 7.** *$i^{th}$ state of a Markov process over $\Omega$ state space is called absorbing if it is impossible to leave it. Actually from $n \geq n_0$ in which $n_0$ is an arbitrary instance of time, the probability of $i^{th}$ state is 1. Consequently, the probability vector will converge to $[0, 1, 0]$ unit vector.*

**Assumption 2.** *Let the internal environment be a stationary environment, then there exists $n^*$ such that for $n \geq n^* \geq \Delta$, $\psi_{int-Stop}$ will be 0.*

**Definition 8.** *Consider a discrete-time system with the following definition:*

$$x(n+1) = f(x(n)) \tag{15}$$

*Here, $x$ and $f$ are vectors, $f(0) = 0$, and $x \neq 0$. Suppose a continuous scalar function $V(x)$ exists that satisfies the following conditions:*

- *$V(x) > 0$ for all $x \neq 0$*

- *$\Delta V(x) < 0$ for all $x \neq 0$, where $\Delta V(x(n+1)) = V(x(n+1)) - V(x(n)) = V(f(x(n))) - V(x(n))$*

- *$V(0) = 0$*

- $V(x) \rightarrow \infty$ if $||x|| \rightarrow \infty$

*Then, $V(x)$ is a Lyapunov function, and the system is stable asymptotically around $x = 0$.*

**Definition 9.** *The function in the definition 8 is said to be contraction if:*

$$||f(x)|| < ||x|| \quad \& \quad f(0) = 0 \tag{16}$$

*Given a non-zero set of values $x$ and a specific norm, the system described above is stable asymptotically. Furthermore, the Lyapunov function of this system is given by:*

$$V(x) = ||x|| \tag{17}$$

**Lemma 2.** *Assume that the internal environment is a stationary environment which is defined in the assumption 2, then the RL agent will converge to the 'Stop' action.*

*Proof.* For this proof, we consider VASLA (detailed definition in Appendix C.2). However, since our model is generalizable, other forms of reinforcement learning agents such as multi-armed bandits and Q-learning can also be applied. The comprehensive proof is provided in Appendix E.1. □

**Lemma 3.** *If the RL agent, using the defined $R_{RL\,Agent}$, converges to the unit vector $[0, 1, 0]$ (indicating 0 probability for both 'Grow' and 'Shrink' actions, and full probability for the 'Stop' action), the entropy will approach 0.*

*Proof.* This lemma has been proved through the definition of entropy. See Appendix E.2. □

**Remark 1.** *Crucially, the RL agent's convergence to the 'Stop' action signifies the discovery of the appropriate TA depth, indicated as $\acute{N}$.*

## 2.4 EXTERNAL ENVIRONMENT ANALYSIS

Before beginning the analysis of the external environment, we should wrap up what has been done so far. We know that $n \geq n^* \geq \Delta$, as a result, the RL agent converges to the 'Stop' action (Assumption 2 and Lemma 2). On the other hand, this convergence leads to the choosing the constant depth $\acute{N}$ for the TA (Remark 1).

**Definition 10.** *External environment is an outer environment with which the TA will interact. The following set of penalty constants, which have a constant value between 0 and 1 for actions $a_1$ to $a_K$, represents the external environment.*

$$C_{ext} = \{c_{ext-1}, c_{ext-2}, ..., c_{ext-K}\} \tag{18}$$

**Proposition 1.** *It is obvious that the probability of being rewarded ($d$) is the complement of the penalty probability ($c$). Hence, $d_{ext-j} = 1 - c_{ext-j}$.*

**Theorem 1.** *If $n \geq n^* \geq \Delta$ and the RL agent, using the defined $R_{RL\,Agent}$, has converged to the 'Stop' action, then the ADTA, with two actions and a constant depth of $\acute{N}$, will be able to learn.*

*Proof.* To prove this theorem, we leverage the steady-state analysis of the Markov chain associated with the transition matrix of the ADTA. This approach is particularly effective when the ADTA reaches a stable depth $\acute{N}$, allowing us to apply steady-state Markov analysis. For a detailed explanation of the proof, see Appendix E.3. □

# 3 EXPERIMENTS

We present numerical simulations to complement and validate our theoretical findings, comparing the ADTA with state-of-the-art methods in the field of TA. We explore three distinct environments representative of many real-world scenarios, including both stationary and non-stationary settings (see Appendix Section F). Synthetic data from stationary environments is used to test our learning algorithm from both internal and external perspectives. Additionally, we evaluate ADTA with various RL agents in Appendix Section G. To further demonstrate ADTA's practical effectiveness, we apply it to two real-world domains: the dropout problem in deep neural networks and blockchain systems. The detailed results of these applications are provided in Appendix H and I, respectively.

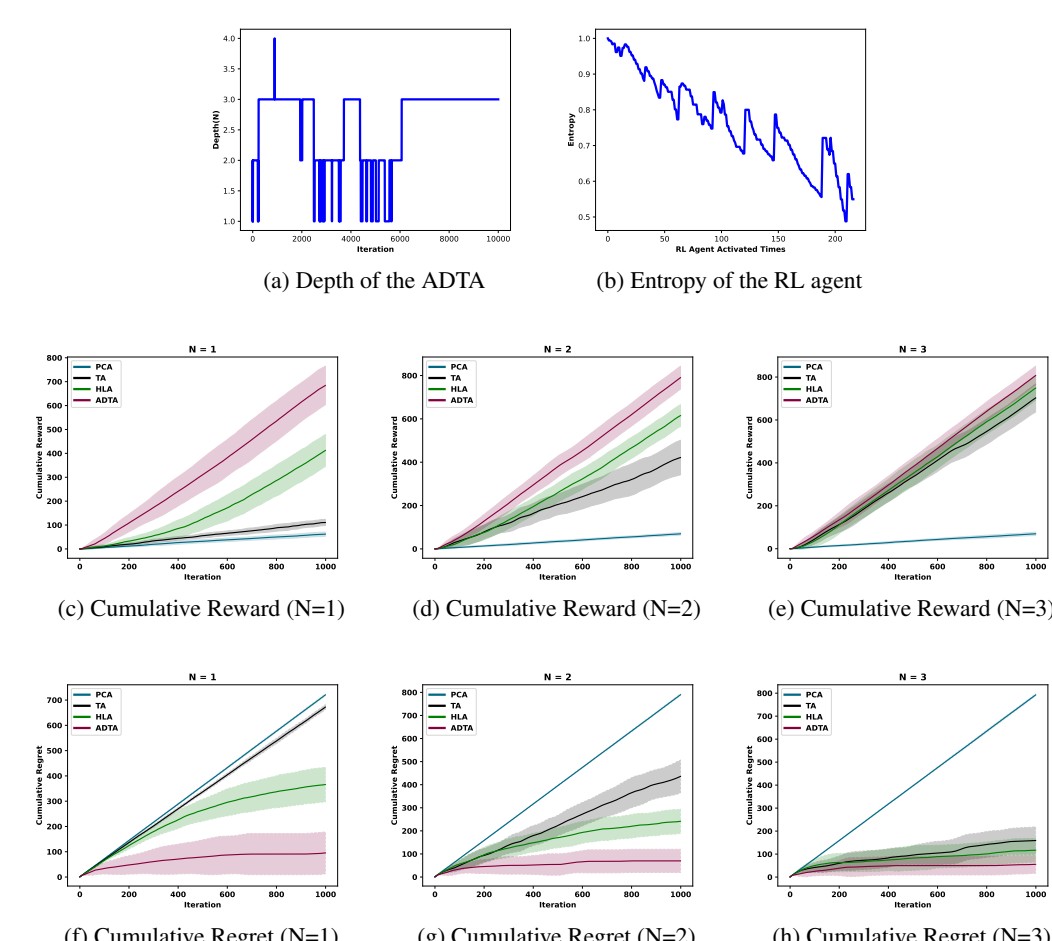

Figure 4: Experimental results of the ADTA considering both internal and external environments

## 3.1 INTERNAL ENVIRONMENT

To explore how the RL agent sets the depth of ADTA, we designed a synthetic environment where we deployed VASLA as the RL agent (detailed explanation in Appendix C.2). This experiment was conducted 10000 times with initial values of $N = 1$. Figure 4a illustrates the exploration and exploitation of the RL agent in finding an appropriate depth. As per Assumptions 1 and 2, around iteration 6000, the RL agent identifies $n^*$ and $\Delta$ (Assumptions 1 and 2), indicating convergence to the 'Stop' action (Lemma 2) and starting point for exploitation. The probability vector of the RL agent is $[0.10, 0.82, 0.08]$. Notably, as suggested by Remark 1, the RL agent identifies $\hat{N} = 3$. This behavior aligns with our theoretical findings in Section 2.3. Finally, Figure 4b illustrates the entropy of the RL agent throughout 10000 iterations of ADTA, with the RL agent activated approximately 200 times for exploration purposes. As exploration increases, the entropy decreases until it reaches zero, confirming its decline as described in Lemma 3.

## 3.2 EXTERNAL ENVIRONMENT

This experiment validates our theoretical findings from Section 2.4 by comparing our approach with a Pure Chance Automaton (PCA, Definition 2) and benchmarking it against the base TA and HLA, which is considered state-of-the-art. The presented results are averages of 20 runs, each with 1000 iterations, using a relatively high number of actions ($K = 50$) and initial $N$ values set to 1, 2, and 3, respectively. In the environment, one action is randomly dominant, with its probability drawn from a

Normal distribution $\mathcal{N}(0.8, 0.05)$, while the other actions are drawn from $\mathcal{N}(0.05, 0.02)$. Through comprehensive evaluation, a depth of three is found to be optimal for this experiment. Figures 4c to 4h demonstrate that ADTA outperforms PCA in terms of both cumulative reward and cumulative regret, supporting our theoretical conclusions in Theorem 1. Additionally, at $N = 1$ and $N = 2$ (Figures 4c, 4d, 4f, and 4g), TA and HLA exhibit excessive exploration due to their fixed depths, while ADTA adapts dynamically to $N = 3$, thereby avoiding over-exploration. As shown in Figures 4e and 4h, since $N = 3$ represents the appropriate depth, all three learning agents perform similarly at this level.

## 4 DISCUSSION

We consider the depth parameter, $N$, in the TA to develop a new learning agent, ADTA, capable of adapting to a wide range of environments, including both stationary and non-stationary scenarios. The ADTA can be analyzed from multiple perspectives, as outlined below.

Regarding the number of iterations, the ADTA inherently requires more iterations compared to standard TA due to its adaptive mechanism designed to balance exploration and exploitation. Unlike TA, which operates with a fixed depth and may continue doing so even when inappropriate, the ADTA dynamically adjusts its depth, reducing the risk of suboptimal performance.

As demonstrated in our experiments, ADTA's performance is particularly prominent at lower depths. An inappropriate depth setting leads to excessive exploration, causing the learning agent to receive fewer rewards. However, ADTA's integrated reinforcement learning mechanism effectively tunes the depth, allowing the agent to converge on the appropriate depth value. At higher depths, the performance gap between ADTA and other learning automaton agents narrows, as the system naturally approaches a near-optimal depth configuration.

Our approach leverages a large number of actions to showcase the scalability of ADTA, setting it apart from traditional learning agents within the learning automaton family. This highlights ADTA's ability to adapt to more complex and dynamic environments effectively.

The inclusion of a reinforcement learning agent, such as VASLA or multi-armed bandits, introduces minimal structural changes to the TA, requiring only the addition of a vector to track the probabilities for 'Grow', 'Stop', and 'Shrink' actions.

The model is highly flexible in terms of RL agent selection. Multiple RL agents, including multi-armed bandits and Q-learning, can be utilized as depth controllers. However, employing these agents may introduce additional parameters, such as $\lambda_1$ and $\lambda_2$ in VASLA or the $\varepsilon$ parameter in the $\varepsilon$-greedy multi-armed bandit, adding a layer of complexity to the learning process.

## 5 CONCLUSION

In this paper, we introduce the *ADTA* as a solution to the explore-exploit dilemma inherent in traditional TA approaches. This dilemma arises from the challenge of selecting an optimal depth for the TA. The ADTA addresses this issue by autonomously adjusting its depth in unknown environments through integration with an RL agent. Leveraging Lyapunov stability theorem and Markov chain processes, we investigate ADTA's learning capabilities. Our comprehensive evaluations consistently demonstrate ADTA's superiority over traditional TA methods and state-of-the-art techniques like HLA. Interesting future research directions include providing stronger proof for ADTA's effectiveness in complex scenarios, exploring the integration of ADTA with HLA to jointly address depth and action selection policies, and investigating asymmetric depth adjustments for the ADTA.

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

# A NOTATIONS

Table 1: Table of notations

| Notation | Description |
|---|---|
| **Main Text and ADTA's Proof (Section E)** | |
| $n$ | The iteration number |
| $K$ | The number of actions |
| $N$ | The depth or the number nodes in each action of TA |
| $\phi_{(i,j)}$ | The $j^{th}$ node of TA's $i^{th}$ action |
| $\phi_{(i,j)} \rightarrow \phi_{(a,b)}$ | The transition from node $\phi_{(i,j)}$ to node $\phi_{(a,b)}$ |
| $A$ | The action set of ADTA |
| $R$ | The reward of ADTA from the environment |
| $A_T$ | The chosen action of TA in ADTA |
| $R_T$ | The reward of TA in ADTA |
| $A_A$ | The chosen action of the RL agent in ADTA |
| $R_A$ | The reward of the RL agent from the environment |
| $M(n)$ | The average penalty of the TA |
| $c_i$ | The penalty of environment to the $i^{th}$ action |
| $\Delta$ | A threshold in which the property of the internal environment can be analyzed as a stationary environment |
| $\psi$ | The constant value of penalty in the internal environment for $n \geq \Delta$ |
| $\Omega$ | The state space of a Markov process |
| $n^*$ | The convergence point of the internal environment to the 'Stop' action |
| $V(x)$ | The Lyapunov function |
| $p(n)$ | The probability vector of a learning automaton |
| $\lambda_1$ | The reward rate of VSLA |
| $\lambda_2$ | The penalty rate of inner VSLA |
| $\mu_i(n)$ | The expected value of the probability for each action |
| $\rho_{ij}(n)$ | $cov(p_i(n), p_j(n))$ |
| **Non-stationary Environments (Section F)** | |
| $E_i$ | Each stationary part of a non-stationary environment |
| $T$ | The transition matrix of a Markov chain |
| $R$ | The reward matrix of a Markov chain |
| $\zeta$ | The increment value of the penalty probability in the State-dependent environment |
| $\chi$ | The decrement value of the penalty probability in the State-dependent environment |
| **Application : Blockchain Security (Section I)** | |
| $L$ | Length of a branch in a fork |
| $W$ | Weight of a branch in a fork |
| $\delta$ | Fail-safe parameter |
| $\delta_{min}$ | The minimum value of the fail-safe parameter |
| $\delta_{max}$ | The maximum value of the fail-safe parameter |
| $\tau$ | Decision-making time |
| $\theta$ | Time Window parameter |

# B ADDITIONAL RELATED WORKS

Reinforcement learning, a pivotal paradigm in machine learning, can be categorized into two main types (Sutton & Barto, 2018): model-based (Moerland et al., 2023; Yi et al., 2018) and model-free (Çalışır & Pehlivanoğlu, 2019; Ramírez et al., 2022; Liu et al., 2021) approaches (Figure 5). In model-based RL, agents construct an internal model of the environment to plan and optimize actions. In contrast, model-free RL involves agents learning directly from interactions with the environment, refining actions through trial-and-error. The choice between these approaches depends on task characteristics, with model-based methods suited for environments where a reliable model is available, and model-free methods excelling in complex and uncertain environments. In this paper, we chose learning automaton from the model-free category.

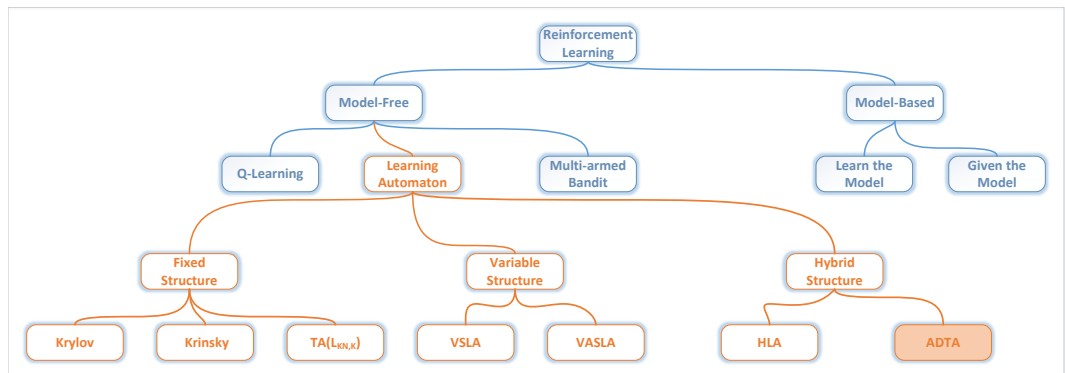

Figure 5: The taxonomy of reinforcement learning algorithms based on the model

**Fixed Structure**. The first class of the learning automaton family is fixed structure. The domain of fixed structure learning automaton (Rezvanian et al., 2018; Zhang & Zhou, 2023) is expansive, encompassing various branches such as Tsetlin ($L_{KN,K}$) (Granmo, 2018; Seraj et al., 2022; Abeyrathna et al., 2021b), Krinsky (Meybodi & Beigy, 2002; Jamalian & Mehrabi, 2022), and Krylov (Khojasteh & Meybodi, 2005). Each learning automaton is dedicated to specific decision-making paradigms, aligning with diverse cognitive abilities observed in human behavior. While the Tsetlin Automaton incorporates rewards and penalties, other models explore cognitive aspects such as impulsivity and greed.

**Variable Structure**. Variable Structure Learning Automaton (Narendra & Thathachar, 2012; Yazidi et al., 2020; Safara et al., 2020; Ghanavati et al., 2020), belonging to the second class of learning automaton, exhibits various types, with VSLA and VASLA being the most significant branches. In VSLA, interaction with the surrounding environment updates a probability vector. VASLA, similar to VSLA, differs in that, in certain situations, not all actions are accessible, resulting in a variable number of actions based on the prevailing circumstances.

**Hybrid Structure**. Representing a notable departure from prior literature, Gholami et al. introduced a significant modification to the TA (Gholami et al., 2023) by combining the fixed structure and variable structure families, thereby establishing the third class and the state-of-the-art family of learning automaton. This alteration specifically targeted the optimization of action switching by learning the best next action.

## C   More About VSLA Family

In this section, we expand upon the concept of the variable structure family of learning automata, as its utilization is integral to various sections of our paper.

### C.1   Variable Structure Learning Automaton (VSLA)

Variable structure learning automaton (Narendra & Thathachar, 2012; Yazidi et al., 2020; Safara et al., 2020; Ghanavati et al., 2020) can be defined mathematically by a quintuple $< A, R, P, \lambda_1, \lambda_2 >$, where $A = \{a_1, a_2, \cdots, a_K\}$ denotes the finite action set from which the automaton can select the intended action, $R$ denotes the reward of the environment ($R \in [0,1]$), $P = \{p_1, p_2, \cdots, p_K\}$ denotes the action probability vector, such that $p_i$ is the probability of choosing the $a_i$ action ($1 \leq i \leq K$), $\lambda_1$, and $\lambda_2$ indicate the reward and penalty parameters that determine the amount of increase and decrease of the action probabilities at epoch $n$.

$\lambda_1$ and $\lambda_2$ can have different values. Based on these values, the updating rule of the probability vector can be categorized as follows:

- $L_{R-P}$: This updating scheme, which is called "linear reward-penalty," comes from the equality of the reward and penalty parameters ($\lambda_1 = \lambda_2$). When both are the same, the probability vector of the learning automaton increases or decreases at a monotonic rate.

- $L_{R-\varepsilon P}$: This updating scheme, which is called "linear reward-$\varepsilon$ penalty," leads to a much greater value of the reward parameter in relation to the penalty parameter ($\lambda_1 >> \lambda_2$).

- $L_{R-I}$: When there is no penalty in an updating scheme ($0 < \lambda_1 < 1, \lambda_2 = 0$), this updating scheme is called "linear reward-Inaction." The probability vector of the learning automaton will not change upon receiving an unfavorable response from the environment.

- $L_{P-I}$: If the conducted probability vector in the learning automaton doesn't change by receiving the favorable action, this updating scheme is called "linear penalty-Inaction" ($\lambda_1 = 0, 0 < \lambda_2 < 1$).

- **Pure Chance**: An updating scheme in which there is no penalty and reward parameter ($\lambda_1 = \lambda_2 = 0$) is called "Pure Chance." In this updating scheme, the probability vector of the automaton will not change in any conditions.

The automaton performs its chosen action on the environment at epoch $n$. If the learning automaton chooses its intended action ($i = j$), the probability vector will update using (19).

$$p_i(n+1) = p_i(n) + \lambda_1(1 - R(n))(1 - p_i(n)) - \lambda_2 R(n) p_i(n) \tag{19}$$

On the other hand, the probability vector for the other actions ($i \neq j$) that are not chosen will update due to the (20).

$$p_j(n+1) = p_j(n) - \lambda_1(1 - R(n)) p_j(n) + R(n)\left(\frac{\lambda_2}{K-1} - \lambda_2 p_j(n)\right) \tag{20}$$

## C.2 Variable Action Set Learning Automaton (VASLA)

Under some circumstances, the number of available actions of the learning automaton varies at each instant. To overcome this constraint, a subset of the variable structure learning automaton called the variable action set learning automaton Thathachar & Harita (1987); Narendra & Thathachar (2012) is defined. Like the variable structure, this automaton can be formulated by a quintuple $< A, R, P, \lambda_1, \lambda_2 >$, where $A = \{a_1, a_2, \cdots, a_K\}$ denotes the finite action set from which the automaton can select the intended action, $R$ denotes the reward of the environment ($R \in [0, 1]$), $P = \{p_1, p_2, \cdots, p_K\}$ denotes the action probability vector, such that $p_i$ is the probability of choosing the $a_i$ action ($1 \leq i \leq K$), $\lambda_1$, and $\lambda_2$ indicate the reward and penalty parameters that determine the amount of increase and decrease of the action probabilities at epoch $n$. Various values for $\lambda_1$ and $\lambda_2$ are defined in Section C.1.

At each epoch, the action subset $\hat{A} \subseteq A$ is available for the learning automaton to choose from. Let $U(n) = \sum_{A_i \in \hat{A}(n)} P_i(n)$ present the sum of probabilities of the available actions in subset $\hat{A}$. Before choosing an action, the available action probability vector is scaled using the (21).

$$\hat{P}_i(n) = \frac{p_i(n)}{U(n)} \quad \forall a_i \tag{21}$$

If the learning automaton chooses its intended action ($i = j$), the probability vector will update using (22).

$$p_i(n+1) = p_i(n) + \lambda_1 R(n)(1 - p_i(n)) - \lambda_2(1 - R(n)) p_i(n) \tag{22}$$

Conversely, the probability vector for the other actions ($i \neq j$) that are not chosen will update due to the (23).

$$p_j(n+1) = p_j(n) - \lambda_1 R(n) p_j(n) + \lambda_2(1 - R(n))\left[\frac{1}{K-1} - p_j(n)\right] \tag{23}$$

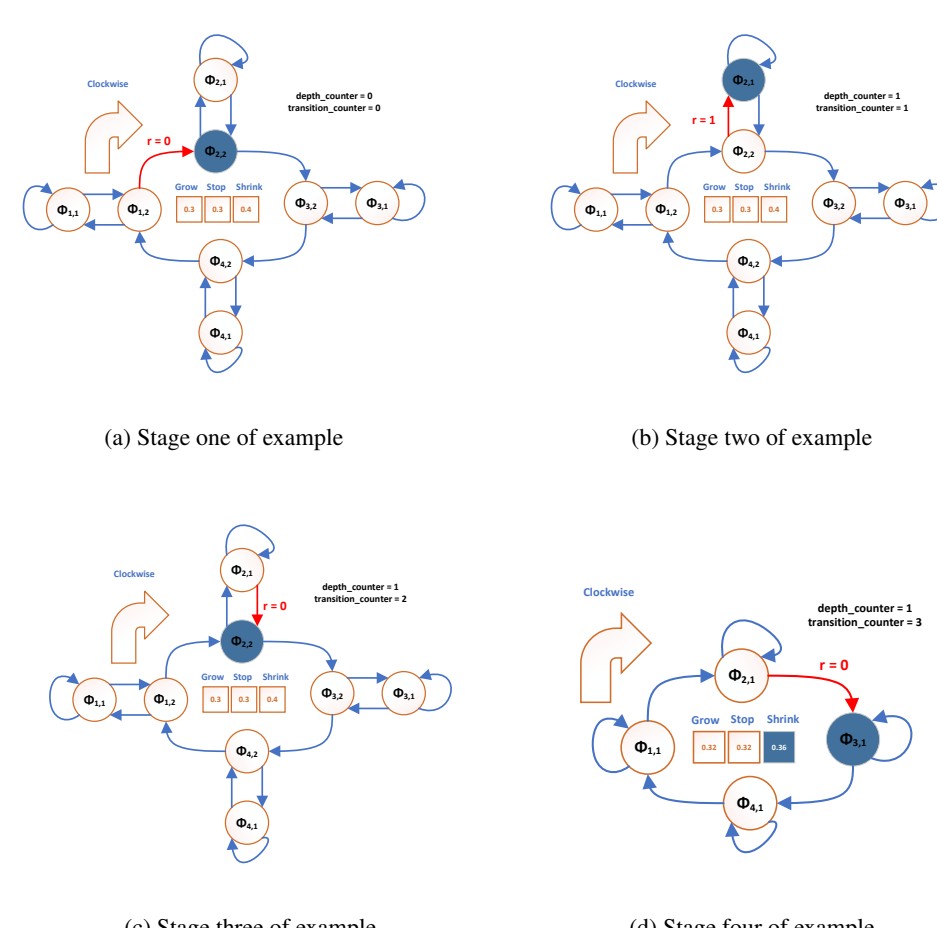

(a) Stage one of example        (b) Stage two of example

(c) Stage three of example        (d) Stage four of example

Figure 6: Four stages of depth decrement

## D NUMERICAL EXAMPLE

This section presents two numerical examples to illustrate the effectiveness of the proposed learning automaton. In these examples, we consider an ADTA with four actions ($K = 4$) and a depth of two ($N = 2$). The RL agent employed is an $L_{R-P}$ VASLA, with $\lambda_1 = \lambda_2 = 0.5$. The first example (Section D.1) demonstrates the depth decrement of the ADTA, while the second example (Section D.2) showcases its depth increment.

### D.1 DEPTH DECREMENT

In this example, illustrated through four stages in Figure 6, after running the proposed learning automaton for a certain period in the environment, the probability vector of the VASLA converges to $[0.3, 0.3, 0.4]$. This indicates a 30% probability for the 'Grow' action, 30% for the 'Stop' action, and 40% for the 'Shrink' action. The resulting probability vector suggests that the ADTA is inclined to reduce its depth.

Initially, let's assume the ADTA switches its action to action number 2, positioning itself at node $\phi_{(2,2)}$, as depicted in Figure 6a. It is important to note that two key variables, *transition_counter* and *depth_counter*, are reset to 0 at this starting point.

In the second stage, the ADTA performs action number 2. This action is successful, and the learning automaton receives a reward from the environment. This reward affects both the *transition_counter*

and *depth_counter* variables. Since the reward causes the ADTA to move to the inner node $\phi_{(2,1)}$, and this transition is a depth transition, the *depth_counter* variable increases by one. Additionally, because a transition occurs, the *transition_counter* also increases by one. This process is illustrated in Figure 6b.

In the third stage, the ADTA performs action 2 again. Unlike the previous stage, the environment deems this action unsuccessful, necessitating a penalty for the ADTA. As a result, the ADTA moves backward to node $\phi_{(2,2)}$. This transition is not a depth transition, so the *depth_counter* remains unchanged. However, since a transition does occur, the *transition_counter* increases by one. Figure 6c illustrates the outcome of the third stage.

In the final stage, the ADTA performs action 2 once more. Similar to the previous stage, the chosen action is unsuccessful, resulting in a penalty. Since the ADTA is at an outer node, it will switch its action. According to the clockwise policy of the ADTA, it will select action 3 and move to the first node of action 3. There are no transitions to a depth node at action 2, so the *depth_counter* remains unchanged. However, the transition to new action will increase the *transition_counter* by one.

In the ADTA, changing the action triggers the activation of VASLA to update the depth of the inner TA. Initially, the VASLA receives feedback regarding its previous depth selection. The feedback $R$ for VASLA is calculated as $\frac{depth\_counter=1}{transition\_counter=3}$, indicating that the previous action of VASLA was moderately effective. Subsequently, the VASLA uses this feedback to update its probability vector for selecting the next action. In this instance, it opts for the 'Shrink' action to decrease the depth, as depicted in Figure 6d, marking the final stage.

The process continues iteratively until the ADTA determines the appropriate depth that accommodates both the environment and its evolving conditions.

## D.2 DEPTH INCREMENT

In this example, which consists of three stages illustrated in Figure 7, the VASLA converges to the probability vector $[0.8, 0.1, 0.1]$, indicating an 80% probability for the 'Grow' action, and 10% each for the 'Stop' and 'Shrink' actions. As the probability vector shows, the VASLA tends to favor the 'Grow' action.

At the first stage, the ADTA is positioned at node $\phi_{(4,1)}$ for the first time, initiating action 4, as depicted in Figure 7a. Since this is the first encounter, both the *transition_counter* and *depth_counter* variables are set to 0.

In the second stage, the ADTA performs action 2, which is rewarded by the environment. This reward returns the ADTA to node $\phi_{(4,1)}$ once more. However, since this transition involves a depth transition, both the *transition_counter* and *depth_counter* variables are incremented by one. This is depicted in Figure 7b.

In the final stage, the ADTA performs action 2 again, but this time receives a penalty from the environment. As a result, it should change its action. The transition from action 4 to action 1 triggers the VASLA, which then receives feedback regarding its prior depth selection. The feedback $R$ is calculated as $\frac{depth\_counter=1}{transition\_counter=2}$. Using this feedback, the VASLA updates its probability vector for selecting the next action, resulting in the new vector $[0.66, 0.17, 0.17]$. In this case, the VASLA chooses the 'Grow' action, increasing the depth from 1 to 2, as shown in Figure 7c.

Additionally, since the ADTA was positioned at the edge node $\phi_{(4,1)}$, it transitions to the next action following the clockwise policy, moving to node $\phi_{(1,2)}$. As this transition does not involve a depth change, the *depth_counter* remains unchanged, while the *transition_counter* is incremented by one.

# E MISSING PROOFS

## E.1 PROOF OF THE RL AGENT CONVERGENCE USING VASLA

Before exploring the proof, we highly recommend familiarizing yourself with the VSLA family, as it is further explained in Section C.

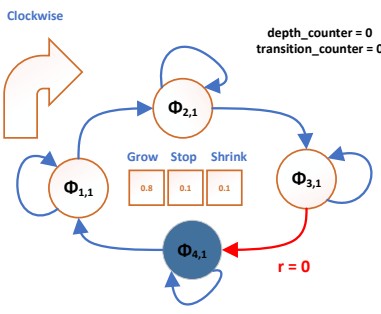

(a) Stage one of example

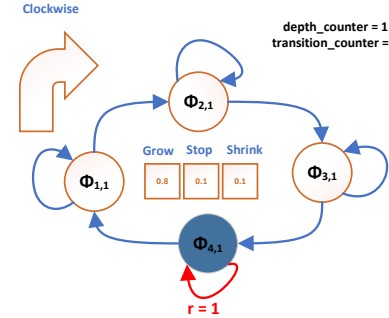

(b) Stage two of example

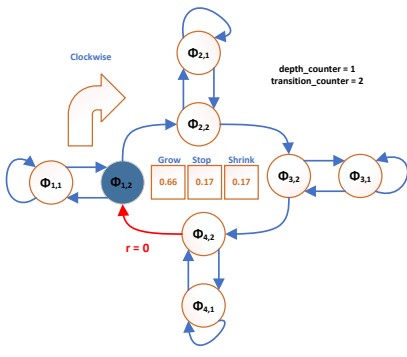

(c) Stage three of example

Figure 7: Three stages of depth increment

*Proof.* To demonstrate the convergence of VASLA to the desired action, we begin by considering the general case where $K$ actions are defined. Subsequently, we focus specifically on VASLA with three actions. The proof initiates by examining the conditional expectation of selecting the desired action (the $i^{th}$ action) as expressed in (24).

$$E[p_i(n+1)|p_i(n)] = \sum_{j=1}^{K} E[p_i(n+1)|p_i(n) \wedge A(n) = a_j]p_j(n) \tag{24}$$

Looking at the main updating equations of VASLA (22) and (23) and the conditional expectation value in (24) will lead us to (25) as a piece-wise function.

$$E[p_i(n+1)|p_i(n)] = \begin{cases} [(1-c_i(n)p_i(n))][p_i(n)+\lambda_1(1-p_i(n))] & R(n)=1, A(n)=a_i \\ [c_i(n)p_i(n)][p_i(n)] & R(n)=0, A(n)=a_i \\ \sum_{j\neq i}^{r}(1-c_j(n))(1-\lambda_1)p_j(n)p_i(n) & R(n)=1, A(n)=a_j \\ \sum_{j\neq i}^{r}c_j(n)p_j(n)p_i(n) & R(n)=0, A(n)=a_j \end{cases} \tag{25}$$

Summation of the above conditional expected value expresses the desired one like (26).

$$E[p_i(n+1)|p_i(n)] = [(1-c_i(n)p_i(n))][p_i(n)+\lambda_1(1-p_i(n))] \tag{26}$$

$$+[c_i(n)p_i(n)][p_i(n)] + \sum_{j\neq i}^{K}(1-c_j(n))(1-\lambda_1)p_j(n)p_i(n)$$

$$+\sum_{j\neq i}^{K}c_j(n)p_j(n)p_i(n)$$

Taking the expected value from both sides leads to (27).

$$E[p_i(n+1)] = \lambda_1 \sum_{j=1}^{K}c_j(n)E[p_i(n)p_j(n)] + (1-\lambda_1 c_i(n))E[p_i(n)] \tag{27}$$

If $E[p_i(n)p_j(n)]$ is substituted with the corresponding covariance term, and for the sake of simplification, $cov(p_i(n),p_j(n)) = \rho_{ij}(n)$ and $E[p_i(n)] = \mu_i(n)$, we will have (28).

$$E[p_i(n+1)] = \lambda_1 \sum_{j=1}^{K}c_j(n)[\rho_{ij}(n)+\mu_j(n)]] + (1-\lambda_1 c_i(n))E[p_i(n)] \tag{28}$$

Now, it is time to be specific about the number of actions. In our VASLA model, we consider three actions: 'Grow', 'Stop', and 'Shrink'. According to Lemma 1, the penalty probabilities $c_{Grow}(n)$, $c_{Stop}(n)$, and $c_{Shrink}(n)$ converge to the constant values $\psi_{int-Grow}$, $\psi_{int-Stop}$, and $\psi_{int-Shrink}$ respectively. Moreover, since $\psi_{int-Stop} = 0$, equations (29) and (30) can be derived directly from the general form of the conditional expected value.

$$\mu_{Grow}(n+1) = \lambda_1 \psi_{int-Grow}\rho_{Grow,Grow}(n) \tag{29}$$

$$+\lambda_1 \psi_{int-Grow}\mu_{Grow}(n)^2 + \lambda_1 \psi_{int-Shrink}\rho_{Grow,Shrink}(n)$$

$$+\lambda_1 \psi_{int-Shrink}\mu_{Grow}(n)\mu_{Shrink}(n) + \mu_{Grow}(n)(1-\lambda_1 \psi_{int-Grow})$$

$$\mu_{Shrink}(n+1) = \lambda_1 \psi_{int-Grow}\rho_{Shrink,Grow}(n) \tag{30}$$

$$+\lambda_1 \psi_{int-Grow}\mu_{Shrink}(n)\mu_{Grow}(n) + \lambda_1 \psi_{int-Shrink}\rho_{Shrink,Shrink}(n)$$

$$+\lambda_1 \psi_{int-Shrink}\mu_{Shrink}(n)^2 + \mu_{Shrink}(n)(1-\lambda_1 \psi_{int-Shrink})$$

To apply the Lyapunov stability theorem and the contraction mapping theorem in definitions (8) and (9), $\xi$ function is defined with input values of $\mu_{Grow}$ and $\mu_{Shrink}$. To apply the definition 8, we should prove that $\xi(\mu_{Grow},\mu_{Shrink})$ is a contraction using the absolute-value norm(L1 norm). This condition is summarized in (31) and (32).

$$||\xi(\mu_{Grow},\mu_{Shrink})||_1 \overset{?}{<} ||\mu_{Grow}(n)||_1 + ||\mu_{Shrink}(n)||_1 \tag{31}$$

$$||\xi(\mu_{Grow},\mu_{Shrink})||_1 = \mu_{Grow}(n)(1-\lambda_1 \psi_{int-Grow}) \tag{32}$$

$$+\lambda_1 \psi_{int-Grow}[\rho_{Grow,Grow}(n) + \mu_{Grow}(n)^2 + \rho_{Shrink,Grow}(n)$$

$$+\mu_{Shrink}(n)\mu_{Grow}(n)] + \mu_{Shrink}(n)(1-\lambda_1 \psi_{int-Shrink})$$

$$+\lambda_1 \psi_{int-Shrink}[\rho_{\mu_{Shrink}(n),Shrink}(n) + \mu_{Shrink}(n)^2 + \rho_{Grow,Shrink}(n)$$

$$+\mu_{Grow}(n)\mu_{Shrink}(n)]$$

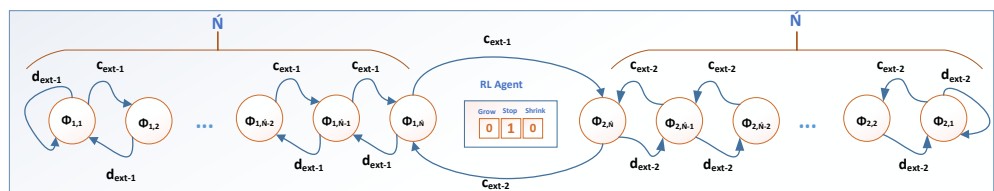

Figure 8: The transition among states in ADTA with 2 actions after the RL agent converges to the 'Stop' action and $\acute{N}$ is chosen for each action

Furthermore, the inequality of (33) can be inferred from the expectation value properties.

$$E[\mu_i(n)^2] + E[\mu_i(n)\mu_j(n)] = E[\mu_i(n)^2 + \mu_i(n)\mu_j(n)] \tag{33}$$
$$= E[\mu_i(n)(\mu_i(n) + \mu_j(n))] < E[\mu_i(n)]$$

Thus, equality of $\xi(\mu_{Grow}, \mu_{Shrink})$ turns into an inequality which is described in (34).

$$||\xi(\mu_{Grow}, \mu_{Shrink})||_1 < \mu_{Grow}(n)(1 - \lambda_1 \psi_{int-Grow}) \tag{34}$$
$$+ \lambda_1 \psi_{int-Grow}\mu_{Grow}(n) + \mu_{Shrink}(n)(1 - \lambda_1 \psi_{int-Shrink})$$
$$+ \lambda_1 \psi_{int-Shrink}\mu_{Shrink}(n)$$

Finally, it is proved that:

$$||\xi(\mu_{Grow}, \mu_{Shrink})||_1 < \mu_{Grow}(n) + \mu_{Shrink}(n) \tag{35}$$

As a result, we can claim that $\xi(\mu_{Grow}, \mu_{Shrink})$ is a contraction mapping function, and due to the definitions 8 and 9, this function will converge to 0 approximately. On the other hand, since the sum of all probabilities equals to 1, the expected value of the 'Stop' action will converge to 1. $\qquad\square$

### E.2 PROOF OF THE RL AGENT ENTROPY

*Proof.* By referring to definition of entropy ($H$), the entropy of the RL agent ($H_{RL\ Agent}$) will be calculated as follows:

$$H = -\sum_{i=1}^{K} p_i \log p_i \tag{36}$$

$$H_{RL\ Agent} = -(p_{Grow} \log p_{Grow} + p_{Stop} \log p_{Stop} + p_{Shrink} \log p_{Shrink}) \tag{37}$$
$$let \quad \lim_{p_{Grow} \to 0} p_{Grow} = \lim_{p_{Shrink} \to 0} p_{Shrink} = 0 \implies$$
$$H_{RL\ Agent} = \lim_{p_{Grow} \to 0, p_{Shrink} \to 0} -[p_{Grow} \log p_{Grow}$$
$$+ (1 - (p_{Grow} + p_{Shrink})) \log(1 - (p_{Grow} + p_{Shrink}))$$
$$+ p_{Shrink} \log p_{Shrink}] \tag{38}$$
$$H_{RL\ Agent} = 0 \tag{39}$$

$\qquad\square$

### E.3 PROOF OF LEARNING CAPABILITY

*Proof.* Now, if the learning ability of the TA is demonstrated, the learning capacity of ADTA will also be established. To achieve this, the average penalty $M(n)$ for the TA must be calculated. Furthermore, based on Definitions 1 to 4, it is essential to determine the probability of each action.

Since the behavior of the TA can be represented using a Markov chain, we employ Markov analysis in the following sections to compute the probability associated with each action.

Considering the transition matrix of the associated TA in Figure 8, we observe that it comprises $2\acute{N} \times 2\acute{N}$ elements, as dictated by the memory size of $2\acute{N}$. Within this matrix, node transitions resulting in a successful reward acquisition are denoted as $d_{ext}$, corresponding to the relevant action. Conversely, if the transition does not lead to success, the element is labeled as $c_{ext}$ based on the action. If neither of these conditions apply, the element is set to 0.

$$T = \left( \begin{array}{cccccc|cccccc}
d_{ext-1} & c_{ext1} & \cdot & \cdot & \cdot & \cdot & \cdot & \cdot & \cdot & \cdot & \cdot & \cdot \\
d_{ext1} & 0 & c_{ext1} & \cdot & \cdot & \cdot & \cdot & \cdot & \cdot & \cdot & \cdot & \cdot \\
0 & d_{ext1} & 0 & c_{ext1} & \cdot & \cdot & \cdot & \cdot & \cdot & \cdot & \cdot & \cdot \\
\vdots & \ddots & & \ddots & & \vdots & \vdots & \ddots & & \ddots & & \vdots \\
0 & 0 & 0 & d_{ext-1} & \cdot & c_{ext-1} & \cdot & \cdot & \cdot & \cdot & \cdot & \cdot \\
\cdot & \cdot & \cdot & \cdot & d_{ext-1} & 0 & \cdot & \cdot & \cdot & \cdot & \cdot & c_{ext-1} \\
\hline
\cdot & \cdot & \cdot & \cdot & \cdot & \cdot & d_{ext-2} & c_{ext-2} & \cdot & \cdot & \cdot & \cdot \\
\cdot & \cdot & \cdot & \cdot & \cdot & \cdot & d_{ext-2} & 0 & c_{ext-2} & \cdot & \cdot & \cdot \\
\cdot & \cdot & \cdot & \cdot & \cdot & \cdot & 0 & d_{ext-2} & 0 & c_{ext-2} & \cdot & \cdot \\
\vdots & \ddots & & \ddots & & \vdots & \vdots & \ddots & & \ddots & & \vdots \\
\cdot & \cdot & \cdot & \cdot & \cdot & \cdot & 0 & 0 & 0 & d_{ext-2} & \cdot & c_{ext-2} \\
\cdot & \cdot & \cdot & \cdot & c_{ext-2} & & \cdot & \cdot & \cdot & \cdot & d_{ext-2} & 0
\end{array} \right)$$

Steady-state calculation of Markov chain with the mentioned transition matrix will yield to the $M(n)$ as follows:

$$M(n) = c_1 p_1 + c_2 p_2$$

$$M(n) = \frac{\frac{1}{c_{ext-1}^{\acute{N}-1}} \times (\frac{c_{ext-1}^{\acute{N}} - d_{ext-1}^{\acute{N}}}{c_{ext-1} - d_{ext-1}}) + \frac{1}{c_{ext-2}^{\acute{N}-1}} \times (\frac{c_{ext-2}^{\acute{N}} - d_{ext-2}^{\acute{N}}}{c_{ext-2} - d_{ext-2}})}{\frac{1}{c_{ext-1}^{\acute{N}}} \times (\frac{c_{ext-1}^{\acute{N}} - d_{ext-1}^{\acute{N}}}{c_{ext-1} - d_{ext-1}}) + \frac{1}{c_{ext-2}^{\acute{N}}} \times (\frac{c_{ext-2}^{\acute{N}} - d_{ext-2}^{\acute{N}}}{c_{ext-2} - d_{ext-2}})}$$

To establish TA's learning capability with 2 actions and $2 \times \acute{N}$ states, we explore three scenarios for $M(n)$ based on $c_{ext-1}$ and $c_{ext-2}$ values: 1) When $c_{ext-1} < c_{ext-2} < \frac{1}{2}$, $c_{ext-1} < d_{ext-1}$ and $c_{ext-2} < d_{ext-2}$, leading to $\lim_{n\to+\infty} M(n)$ reaching 0. Thus, ADTA's $M(n)$ is below $M_0$, indicating its learning capability. 2) For $c_{ext-1} < \frac{1}{2} < c_{ext-2}$, $\lim_{n\to+\infty} M(n)$ converges to $c_1$, which is less than $M_0$, confirming ADTA's learning ability. 3) In cases of $c_{ext-1} > \frac{1}{2}, c_{ext-2} > \frac{1}{2}$, $\lim_{n\to+\infty} M(n)$ is determined by $\frac{4c_{ext-1}c_{ext-2} - c_{ext-1} - c_{ext-2}}{2c_{ext-1} + 2c_{ext-2} - 2}$, emphasizing the influence of $c_{ext-1}$ and $c_{ext-2}$ values on ADTA's learning capability. $\qquad\square$

# F    NON-STATIONARY ENVIRONMENTS EXPERIMENTS

In this section, we present additional synthetic experiments on more complex environments known as non-stationary environments. In these environments, where the penalty probability $c_i$ changes over time, fixed strategies used by learning automaton may become ineffective or result in frequent penalties Narendra & Thathachar (2012). To succeed in such conditions, learning automaton must demonstrate adaptability. Non-stationary environments can be analyzed by dividing them into time intervals with constant penalty probabilities, resembling the process of learning in multiple random environments. We focus on two types of non-stationary environments: Markovian switching and State-dependent. In these cases, the learning automaton operates within a finite set of environments, denoted as $E_1, E_2, ..., E_D$ Narendra & Thathachar (2012). To maintain consistency with other sections of the paper that utilize learning automaton theory, we employ the VASLA as the reinforcement learning agent.

## F.1    MARKOVIAN SWITCHING ENVIRONMENT

In a Markovian switching environment, each environment $E_i; (1 \le i \le D)$ corresponds to a distinct state of a Markov chain. If the chain is ergodic, the learning automaton interacting with this environment will occupy each state with a fixed probability, dictated by the asymptotic probability distribution of the ergodic chain Narendra & Thathachar (2012); Thathachar & Sastry (2003).

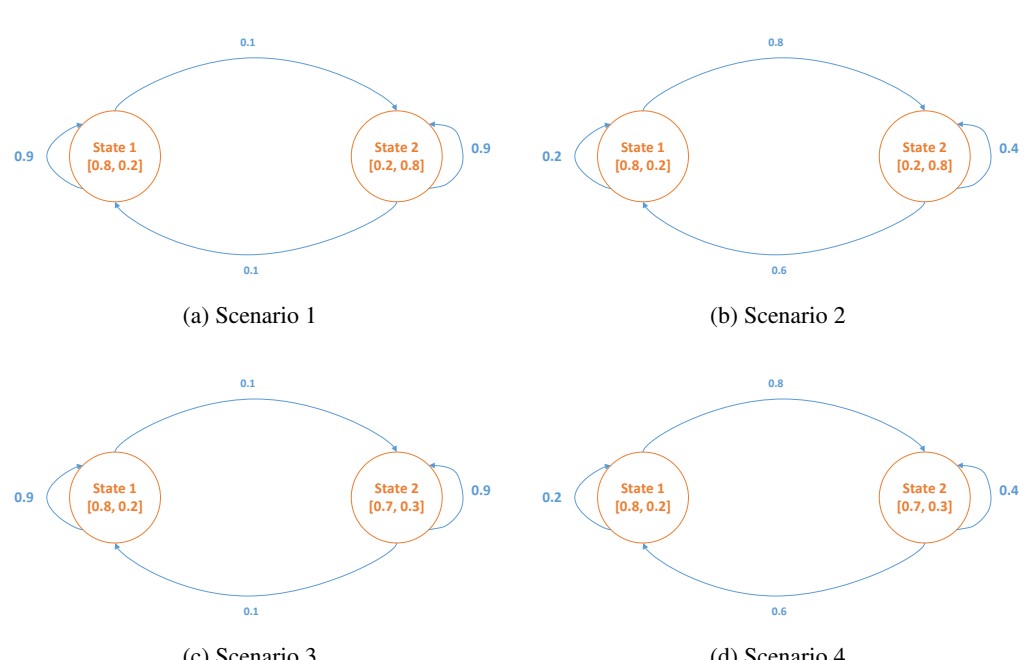

(a) Scenario 1            (b) Scenario 2

(c) Scenario 3            (d) Scenario 4

Figure 9: Four scenarios of Markovian switching environment

### F.1.1 SIMPLE MARKOV CHAIN

These experiments consist of four distinct scenarios, each representing a unique Markovian switching environment governed by a two-state Markov chain. All scenarios are illustrated in Figure 9. The scenarios include the following:

(i) The transition from one state to another changes the action with the higher reward probability, as shown in Figure 9a. The likelihood of remaining in the current state is greater than transitioning between states. Steady-state analysis converts this environment into a stationary one with action probabilities of $[0.5, 0.5]$.

(ii) Transitioning between states completely alters the action with the highest reward probability, with a high likelihood of state changes (Figure 9b). Steady-state analysis converts this environment into a stationary one with action probabilities of $[0.45, 0.54]$.

(iii) State transitions affect the reward probabilities, but the action with the highest reward probability remains unchanged. There is a high tendency to stay in the current state (Figure 9c). Steady-state analysis converts this environment into a stationary one with action probabilities of $[0.75, 0.24]$.

(iv) The action with the highest reward probability remains consistent across states, but state transitions can affect the likelihood of receiving a reward for that action. Changing states is relatively easy (Figure 9d). Steady-state analysis converts this environment into a stationary one with action probabilities of $[0.74, 0.26]$.

In all four scenarios, common configurations were applied: the inner VASLA followed the $L_{R-I}$ model with $\lambda_1 = 0.01$ and $\lambda_2 = 0$; and the initial depth was set to 1, 2, 3, 5, or 7. The reported results are based on 20 realizations, with each realization consisting of 10000 iterations.

The results in Table 2 demonstrate the superiority of the ADTA over the TA and HLA in most experiments. This advantage is due to the adaptive nature of the ADTA in finding an appropriate depth to balance exploration and exploitation. In the first two scenarios (i and ii), the environment is random, so we don't expect the learning automata to perform exceptionally well, as evidenced by achieving around 5000 rewards out of 10000 rounds. However, this changes in scenarios iii and iv,

Table 2: The experimental results of a Simple Markov Chain with respect to cumulative reward

| Model | $N=1$ | $N=2$ | $N=3$ | $N=5$ | $N=7$ |
|---|---|---|---|---|---|
| | | | *Scenario(i)* | | |
| TA | $4974.15 \pm 87.45$ | $4977.35 \pm 83.78$ | $4985.55 \pm 89.01$ | $4997.8 \pm 74.18$ | $4972.55 \pm 65.35$ |
| HLA | $4982.05 \pm 67.43$ | $4997.85 \pm 89.20$ | $4996.6 \pm 71.97$ | $5001.45 \pm 111.68$ | $\mathbf{5008.2 \pm 90.50}$ |
| ADTA | $\mathbf{5016.7 \pm 114.44}$ | $\mathbf{5008.4 \pm 98.83}$ | $\mathbf{5039.45 \pm 107.0}$ | $\mathbf{5030.95 \pm 78.46}$ | $5007.8 \pm 106.47$ |
| | | | *Scenario(ii)* | | |
| TA | $4572.7 \pm 49.47$ | $\mathbf{4578.7 \pm 34.95}$ | $4562.55 \pm 33.45$ | $4549.1 \pm 45.91$ | $4568.3 \pm 29.40$ |
| HLA | $4563.35 \pm 47.94$ | $4566.7 \pm 48.29$ | $4560.95 \pm 38.94$ | $4573.1 \pm 46.22$ | $4544.3 \pm 45.37$ |
| ADTA | $\mathbf{4576.4 \pm 37.97}$ | $4574.9 \pm 38.73$ | $\mathbf{4576.75 \pm 49.98}$ | $\mathbf{4575.3 \pm 47.28}$ | $\mathbf{4578.15 \pm 49.69}$ |
| | | | *Scenario(iii)* | | |
| TA | $7503.8 \pm 49.69$ | $7503.65 \pm 42.99$ | $7505.3 \pm 39.29$ | $7492.75 \pm 44.76$ | $7505.05 \pm 41.46$ |
| HLA | $7505.45 \pm 32.93$ | $7503.6 \pm 43.55$ | $7493.6 \pm 47.33$ | $7510.1 \pm 47.90$ | $7492.85 \pm 35.75$ |
| ADTA | $\mathbf{7512.35 \pm 47.01}$ | $\mathbf{7513.15 \pm 47.36}$ | $\mathbf{7529.85 \pm 35.21}$ | $\mathbf{7519.35 \pm 49.00}$ | $\mathbf{7515.35 \pm 33.17}$ |
| | | | *Scenario(iv)* | | |
| TA | $7430.1 \pm 49.72$ | $7423.95 \pm 34.94$ | $7423.85 \pm 39.89$ | $7420.65 \pm 37.67$ | $7429.3 \pm 36.44$ |
| HLA | $7435.55 \pm 40.93$ | $7415.75 \pm 55.47$ | $7423.6 \pm 38.56$ | $7418.15 \pm 33.98$ | $7423.85 \pm 39.08$ |
| ADTA | $\mathbf{7437.55 \pm 32.15}$ | $\mathbf{7438.9 \pm 42.62}$ | $\mathbf{7445.25 \pm 51.07}$ | $\mathbf{7430.65 \pm 40.01}$ | $\mathbf{7432.1 \pm 33.92}$ |

where one action is dominant, and the LAs, particularly the ADTA, are able to identify this action with a higher reward probability.

F.1.2 COMPLEX MARKOV CHAIN

This experiment aims to evaluate the performance of the proposed learning automaton in a complex Markovian switching environment, focusing on the cumulative reward metric. The environment consists of a Markov chain with four states, as depicted in Figure 10. The transition matrix ($T$) and the reward matrix ($R$), which define the reward probabilities, are as follows:

$$T = \begin{pmatrix} 0.3 & 0.2 & 0.1 & 0.4 \\ 0.1 & 0.2 & 0.5 & 0.2 \\ 0.2 & 0.2 & 0.2 & 0.6 \\ 0.2 & 0.5 & 0.1 & 0.2 \end{pmatrix} \tag{40}$$

$$R = \begin{pmatrix} 0.9 & 0.1 & 0.3 & 0.7 & 0.1 \\ 0.1 & 0.9 & 0.7 & 0.6 & 0.2 \\ 0.3 & 0.7 & 0.5 & 0.5 & 0.3 \\ 0.9 & 0.9 & 0.9 & 0.4 & 0.6 \end{pmatrix} \tag{41}$$

To achieve our goals in this experiment, the inner VASLA adopts the $L_{R-I}$ method with parameters $\lambda_1 = 0.01$ and $\lambda_2 = 0$. Five actions are allowed, and the initial depths considered are $N = 1, 2, 3, 5, 7$.

Before analyzing the results presented in Table 3 for 20 realizations, each consisting of 10000 iterations, the Markovian switching environment is transformed into a stationary environment using steady-state analysis. This conversion yields reward probabilities of $[0.52, 0.68, 0.62, 0.52, 0.31]$ for actions 1 to 5, respectively.

Table 3: The experimental results of a Complex Markov Chain with respect to cumulative reward

| Model | $N=1$ | $N=2$ | $N=3$ | $N=5$ | $N=7$ |
|---|---|---|---|---|---|
| TA | $5821.5 \pm 174.53$ | $6053.95 \pm 53.42$ | $6278.0 \pm 70.35$ | $6536.9 \pm 79.76$ | $6716.85 \pm 118.58$ |
| HLA | $6568.65 \pm 152.50$ | $6538.1 \pm 132.71$ | $6486.65 \pm 112.93$ | $6625.95 \pm 101.96$ | $6748.45 \pm 118.58$ |
| ADTA | $\mathbf{6688.7 \pm 174.53}$ | $\mathbf{6650.25 \pm 208.94}$ | $\mathbf{6701.7 \pm 130.76}$ | $\mathbf{6701.55 \pm 188.26}$ | $\mathbf{6750.75 \pm 159.94}$ |

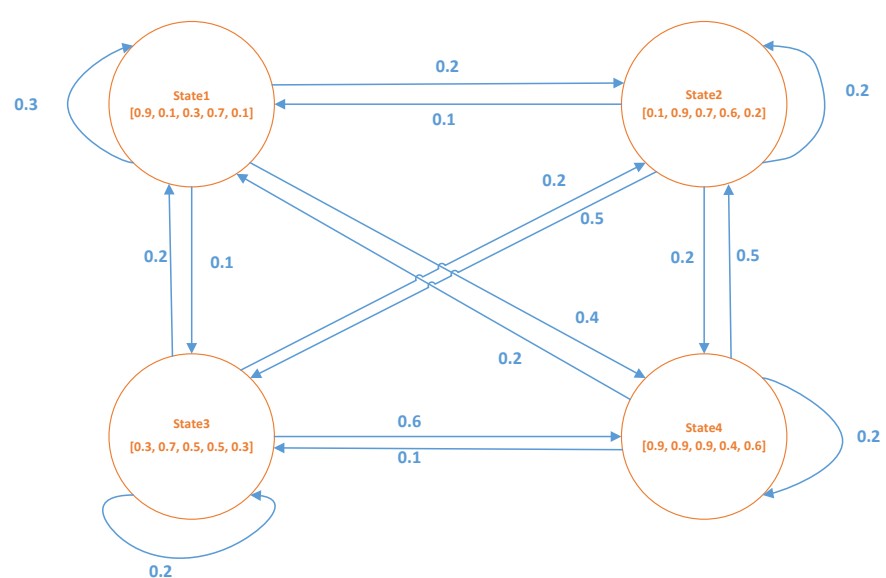

Figure 10: The designed environment for Markovian switching experiment

The ADTA outperforms both the base method (TA) and the state-of-the-art method (HLA) in terms of rewards obtained. This superior performance is attributed to the ADTA's ability to select the appropriate depth. When examining the probability vector after reaching a steady-state, it becomes evident that no single action has a significantly higher probability of being rewarded than the others. In such situations, the ADTA effectively adjusts its depth to maximize its rewards.

## F.2 STATE-DEPENDENT ENVIRONMENT

In a State-dependent environment, when the learning automaton performs action $a_i$ $(1 \leq i \leq K)$ during the $n^{th}$ iteration, the probability $c_i$ associated with that action increases. In contrast, the probabilities of the other actions decrease. Consequently, the performed action becomes less favorable in subsequent stages, whereas the other actions become more advantageous over time Narendra & Thathachar (2012).

Mathematically, this environment is described by the following equations:

$$\begin{cases} c_i(n) = c_i(n) + \zeta_i(n) & i = j \\ c_j(n) = c_j(n) - \chi_j(n) & i \neq j \end{cases} \tag{42}$$

In the given equation, $\zeta_i(n)$ and $\chi_j(n)$ $(i, j = 1, 2, ..., K)$ are constants associated with the $n^{th}$ iteration. The next equation represents the constant value of $\zeta_i(n)$, which will increase the probability of the chosen action:

$$\zeta_i(n) = \begin{cases} \zeta_i & c_i(n) + \zeta_i(n) \leq 1 \\ 1 - c_i(n) & o.w \end{cases} \tag{43}$$

And $\chi_j(n)$ represents a constant value that decreases the probability of other actions, as described by the following equation:

$$\chi_j(n) = \begin{cases} \chi_j & c_j(n) - \chi_j(n) \geq 0 \\ 1 - c_j(n) & o.w \end{cases} \tag{44}$$

### F.3 State-Dependent Experiment

This experiment focuses on assessing the performance of the ADTA in a State-dependent environment, comparing it with the base model (TA) and the state-of-the-art model (HLA) using the cumulative reward metric.

To design the State-dependent environment, three sets of $(\zeta, \chi)$ tuples are considered: $(0.0002, 0.00002)$, $(0.0005, 0.0005)$, and $(0.00002, 0.0002)$ for scenarios 1 to 3, respectively. Additionally, the initial action probabilities in this environment are set to $[0.9, 0.1]$, which can dynamically change based on the values of $\zeta$ and $\chi$.

The inner VASLA adopts the $L_{R-I}$ strategy with $\lambda_1 = 0.01$ and $\lambda_2 = 0$. Initial depths of 1, 2, 3, 5, and 7 are considered for various configurations. The reported results are based on 20 realizations, with each realization consisting of 1000 iterations.

Table 4: The experimental results of a state-dependent environment concerning cumulative reward

| Model | $N = 1$ | $N = 2$ | $N = 3$ | $N = 5$ | $N = 7$ |
|---|---|---|---|---|---|
| | Scenario $1 - (\zeta = 0.0002, \chi = 0.00002)$ | | | | |
| TA | $700.75 \pm 19.01$ | $770.15 \pm 13.35$ | $784.45 \pm 11.90$ | $795.55 \pm 13.59$ | $801.3 \pm 14.69$ |
| HLA | $748.5 \pm 14.36$ | $773.95 \pm 14.94$ | $788.5 \pm 14.46$ | $795.45 \pm 12.18$ | $797.95 \pm 13.20$ |
| ADTA | $796.1 \pm 14.12$ | $799.6 \pm 14.11$ | $794.15 \pm 15.35$ | $800.0 \pm 13.22$ | $803.85 \pm 11.50$ |
| | Scenario $2 - (\zeta = 0.0005, \chi = 0.0005)$ | | | | |
| TA | $637.45 \pm 16.15$ | $653.2 \pm 12.97$ | $657.9 \pm 12.43$ | $656.85 \pm 12.20$ | $655.05 \pm 19.65$ |
| HLA | $655.9 \pm 15.88$ | $656.45 \pm 10.80$ | $657.5 \pm 13.37$ | $657.95 \pm 10.47$ | $657.65 \pm 14.50$ |
| ADTA | $660.2 \pm 16.46$ | $662.4 \pm 16.35$ | $665.35 \pm 13.87$ | $663.75 \pm 14.60$ | $661.45 \pm 10.11$ |
| | Scenario $3 - (\zeta = 0.00002, \chi = 0.0002)$ | | | | |
| TA | $821.25 \pm 13.61$ | $880.6 \pm 10.01$ | $888.5 \pm 9.68$ | $884.0 \pm 9.61$ | $891.6 \pm 10.31$ |
| HLA | $844.35 \pm 12.41$ | $874.85 \pm 13.99$ | $887.05 \pm 8.44$ | $888.65 \pm 9.48$ | $889.85 \pm 10.51$ |
| ADTA | $885.95 \pm 11.10$ | $883.8 \pm 9.46$ | $889.3 \pm 12.66$ | $891.55 \pm 7.62$ | $892.35 \pm 9.92$ |

The results are presented in Table 4. In the first scenario, the probability of being penalized increases by $\zeta = 0.0002$, meaning the optimal action weakens after some iterations while other actions strengthen at a rate of $\chi = 0.00002$. In the second scenario, the weakening of the optimal action occurs at a lower rate, equal to the strengthening rate of other actions ($\zeta = \chi = 0.0002$). In the third scenario, the dominant action is minimally affected ($\zeta = 0.00002$). In all conditions, the ADTA demonstrates superiority over TA and HLA in terms of cumulative reward, attributed to the effective configuration of the depth parameter.

## G Various RL Agents

In most sections of this paper, we use VASLA as the RL agent since the primary focus of this work is on learning automaton. However, in this section, we explore the impact of substituting other multi-armed bandit algorithms Lattimore & Szepesvári (2020); Kalvit & Zeevi (2021) for the RL agent. Specifically, we experiment with UCB-1 Amani & Thrampoulidis (2021), Thompson Sampling Jin et al. (2022), Softmax Elena et al. (2021), and $\varepsilon - greedy$ Hossain et al. (2021) from the multi-armed bandit family to investigate their effects on ADTA's performance.

For this experiment, we consider a stationary environment where one action is randomly dominant, with its probability drawn from a Normal distribution $\mathcal{N}(0.8, 0.05)$, while the other actions are drawn from $\mathcal{N}(0.05, 0.02)$. The ADTA is configured with 20 actions and an initial depth of $N = 1$, making it more challenging for ADTA to identify the appropriate depth.

The results in Figure 11 demonstrate that $\varepsilon - greedy$ outperforms other RL agents in terms of both cumulative reward and cumulative regret. This superior performance can be attributed to its effective balance between exploration and exploitation. While VASLA and Softmax also show relatively

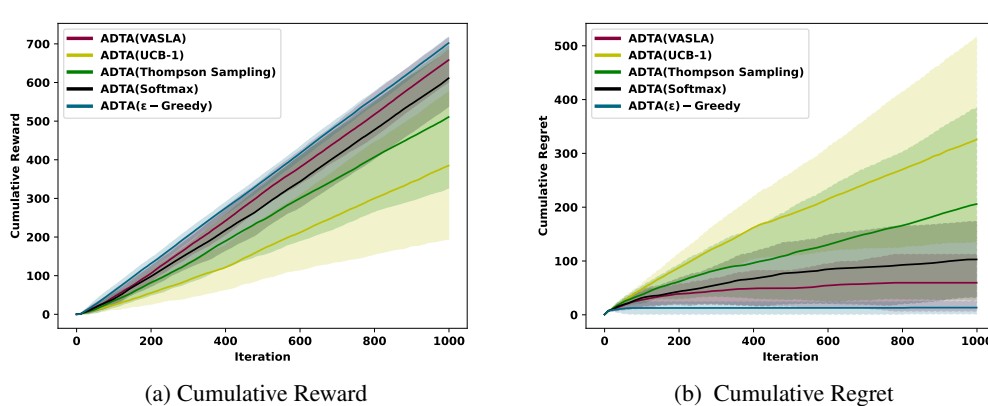

(a) Cumulative Reward                              (b) Cumulative Regret

Figure 11: The experimental results of various RL agents acting as a depth controller

good results, UCB-1 and Thompson Sampling lag behind, likely due to over-exploration or slower adaptation to the environment. These findings highlight the simplicity and balanced nature of $\varepsilon -$ *greedy*, making it an effective agent for the ADTA in this particular scenario.

## H   APPLICATION : DROPOUT TECHNIQUE

Addressing overfitting in deep neural networks, especially in large architectures, presents a formidable challenge. The dropout technique (Srivastava et al., 2014), introduced to mitigate this issue, involves randomly omitting neurons and connections during network training to curb excessive co-adaptation. It employs random unit dropping with a fixed retention probability, typically within the [0.5, 1] range. This fixed probability lacks adaptability and requires extensive experimentation to determine suitable values for various network configurations. (Gholami et al., 2023) pioneered the application of LA to this problem, incorporating a three-action *HLA*, with 'increase,' 'decrease,' and 'stop' actions to adjust dropout probabilities. The HLA avoids exceeding preset bounds. Thinned networks are sampled by the HLA during training, and employed in forward and back-propagation within mini-batches. A single HLA manages dropout probability, adjusted per mini-batch. Gradients are averaged within mini-batches, and LA reinforcement signals depend on thinned network loss values.

Table 5: Dropout Results

| Model | Config 1 | | Config 2 | | Config 3 | | Config 4 | | Config 5 | |
|---|---|---|---|---|---|---|---|---|---|---|
| | Mean | Std | Mean | Std | Mean | Std | Mean | Std | Mean | Std |
| | **N=1** | | | | | | | | | |
| HLA | 0.9579 | 0.0055 | 0.9599 | 0.0073 | 0.9588 | 0.0057 | 0.9598 | 0.0046 | 0.9593 | 0.0065 |
| ADTA | **0.9626** | **0.0036** | **0.9629** | **0.0038** | **0.9619** | **0.0036** | **0.9601** | **0.0049** | **0.9632** | **0.0021** |
| | **N=3** | | | | | | | | | |
| HLA | 0.9603 | 0.0039 | 0.9576 | 0.0075 | 0.9615 | 0.0044 | 0.9591 | 0.0049 | 0.9588 | 0.0059 |
| ADTA | **0.9622** | **0.0046** | **0.9608** | **0.0051** | **0.9620** | **0.0048** | **0.9618** | **0.0041** | **0.9615** | **0.0035** |
| | **N=5** | | | | | | | | | |
| HLA | 0.9552 | 0.0077 | 0.9594 | 0.0073 | 0.9578 | 0.0055 | 0.9620 | 0.0042 | 0.9596 | 0.0068 |
| ADTA | **0.9622** | **0.0049** | **0.9606** | **0.0047** | **0.9619** | **0.0033** | **0.9620** | **0.0030** | **0.9624** | **0.0033** |
| | **N=7** | | | | | | | | | |
| HLA | 0.9568 | 0.0059 | 0.9597 | 0.0063 | 0.9571 | 0.0074 | 0.9599 | 0.0073 | 0.9575 | 0.0071 |
| ADTA | **0.9626** | **0.0033** | **0.9614** | **0.0044** | **0.9630** | **0.0033** | **0.9613** | **0.0055** | **0.9626** | **0.0035** |

ADTA's evaluation involves replacing HLA with ADTA in a feedforward neural network using the MNIST dataset. Initial depths of 1, 3, 5, and 7 were explored along with 5 distinct inner VASLA configurations. Configurations included PCA (config 1 with $\lambda_1 = \lambda_2 = 0$), $L_{R-I}$ (config 2 with $\lambda_1 = 0.01$, $\lambda_2 = 0$), $L_{P-I}$ (config 3 with $\lambda_1 = 0$, $\lambda_2 = 0.01$), $L_{R-P}$ (config 4 with $\lambda_1 = 0.01$, $\lambda_2 = 0.01$), and $L_{R-\varepsilon P}$ (config 5 with $\lambda_1 = 0.1$, $\lambda_2 = 0.01$). The simulation results, depicted in Table 5, present mean accuracy and standard deviation. The findings underline ADTA's performance superiority over HLA, concerning mean accuracy and standard deviation.

## I  APPLICATION : BLOCKCHAIN SECURITY

The Bitcoin network is inherently dynamic, making it challenging to arrive at deterministic decisions. Therefore, a probabilistic decision-making mechanism is essential for critical decision-making in this environment. Given the vast state space, it is more efficient to employ a single-state decision-maker. As a result, we leverage ADTA to design a novel defense mechanism, named Nik Nikhalat-Jahromi et al. (2024; 2023), aimed at countering the selfish mining attack Eyal (2015); Eyal & Sirer (2018) in Bitcoin Nakamoto (2008); Wang et al. (2019); Babaioff et al. (2012).

We begin with a brief introduction to relevant concepts such as Bitcoin mining and selfish mining. Following this, we detail the experiments conducted to evaluate the automaton's performance in this complex setting. Additionally, the developed simulator is available on GitHub[1].

### I.1  CONCEPTS

Bitcoin Nakamoto (2008), introduced by Satoshi Nakamoto in 2009, is a decentralized cryptocurrency that has gained significant attention due to its decentralized nature Wang et al. (2019).

Transactions in the Bitcoin network are recorded in blocks, and creating a new block requires solving a cryptographic puzzle, which comes with a dedicated reward. Participants who contribute resources to solve these puzzles are known as miners Nakamoto (2008); Eyal & Sirer (2018); Nikhalat-Jahromi et al. (2024; 2023); Wang et al. (2019).

The mining process incentivizes the safety of Bitcoin by rewarding miners based on their shared resources, ensuring the network's decentralization Wang et al. (2019).

However, maintaining Bitcoin's decentralization is a challenging task, as attacks like selfish mining Eyal & Sirer (2018) threaten its fundamental properties. Selfish miners keep newly discovered blocks private and reveal them selectively to maximize their rewards.

When selfish miners reveal their withheld blocks, a fork occurs in the blockchain. In such situations, the honest branch of the fork, resulting from valid work, is discarded, and consensus is reached on the selfish branch Wang et al. (2021).

Our proposed automaton presents a novel approach to address this issue in Bitcoin. Our goal is to simplify the problem by making decisions among the forked branches within each distributed miner. This approach aims to overcome the challenges posed by selfish mining and ensure the integrity of the Bitcoin network.

### I.2  PROPOSED DEFENSE

In this section, we introduce our novel defense mechanism that leverages the power of learning automaton to address the challenge of selfish mining in Bitcoin. The learning automaton serves as a decision-maker at each node, assisting in the selection of a branch from the forked blockchain, even in the presence of selfishly mined branches. To make informed decisions, predefined criteria based on branch characteristics are employed:

- **Branch Length ($L$)**: It represents the number of blocks in a specific branch of the fork.
- **Branch Weight ($W$)**: Calculated by comparing the blocks of a branch with the same height in other branches. The branch with the most recent creation time is incremented by one at each iteration.

---

[1]The link has been removed due to the blind review

To facilitate branch selection, the following parameters are taken into consideration:

- **Fail-Safe Parameter** ($\delta$): This parameter helps miners choose a branch based on $L$ or $W$. If the length of a branch in the fork exceeds the others by a threshold of $\delta$, that branch is chosen. Otherwise, the branch with the highest weight, as determined by $W$, is selected.

- **Decision-Making Time** ($\tau$): This refers to the duration a miner takes to check for existing forks and make a decision. If a fork is detected, the miner considers the $\Delta$ parameter for branch selection.

- **Time Window Parameter** ($\theta$): It configures the next value of $\Delta$ using the learning automaton. Each $\theta$ consists of multiple $\tau$ intervals.

The decision-making algorithm for branch selection involves the following steps:

1. Calculation of $L$ for each branch.

2. Calculation of $W$ for each branch.

3. Sorting the branches in descending order based on length. If the difference between the length of the longest and second-longest branch is greater than $\delta$, the longest branch is chosen. Otherwise, the branch with the highest weight is selected.

4. When $\tau$ reaches its end, the learning automaton determines the next value of $\delta$. Typically, $\delta$ oscillates between a minimum value ($\delta_{min}$) and a maximum value ($\delta_{max}$). The learning automaton has three options: 1) "Grow" to increase $\delta$ by one, 2) "Stop" to keep $\delta$ unchanged, and 3) "Shrink" to decrease $\Delta$ by one.

5. When $\theta$ reaches its end, the learning automaton receives feedback from the environment. We have designed a virtual environment to provide information about the learning automaton's decision. The reward ($R$) is computed by dividing the number of decisions made based on $W$ by the total number of decisions, which includes decisions based on length and weight. The following equation demonstrates the $R$ parameter of the learning automaton.

$$R = \frac{Number\ of\ Weight\ Decisions}{Total\ Number\ of\ Decisions} \tag{45}$$

By following these steps, the proposed defense mechanism enables miners to make informed decisions in the presence of selfish mining, ensuring the integrity and security of the Bitcoin network.

## I.3 EVALUATION

The performance evaluation of the learning automaton against the selfish mining attack considers two metrics:

1. **Relative Revenue**: This metric measures a miner's revenue in comparison to others. The calculation is based on the ratio of the number of blocks mined by the $i^{th}$ miner to the total number of mined blocks Eyal & Sirer (2018).

2. **Lower Bound Threshold**: This metric determines the minimum computational power that a selfish miner must possess to initiate an attack Eyal & Sirer (2018).

## I.4 EXPERIMENT

This experiment evaluates the proposed defense mechanism, implemented using the ADTA, in comparison to the well-known tie-breaking defense and previous VASLA-based defense. Tie-breaking Eyal & Sirer (2018) involves miners randomly selecting a branch when encountering a fork. The study examines the effectiveness of the defense from the perspective of selfish miners, who form a separate group and deviate from the honest miners following the standard Bitcoin protocols.

For the experiment, 10000 blocks are generated in each of the 20 runs, with the parameter $\delta$ varying between $\delta_{min} = 1$ and $\delta_{max} = 3$. The type of VASLA used is $L_{R-\varepsilon P}$ with $\lambda_1 = 0.1$ and $\lambda_2 = 0.01$.

The results shown in Figure 12 demonstrate that the ADTA effectively adapts to complex environments like blockchain, even without prior information. This adaptability leads to the proposed

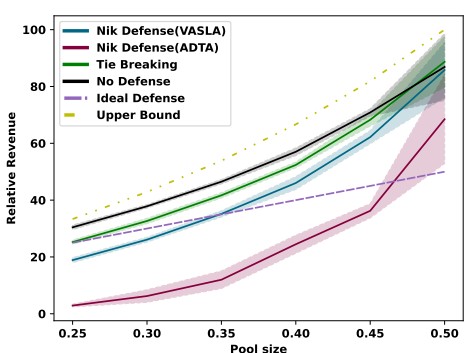

Figure 12: Performance comparison of the proposed defense mechanism against tie-breaking and other learning automaton-based solutions, measured in terms of relative revenue.

defense's superiority over the VASLA-based solution Nikhalat-Jahromi et al. (2024; 2023), as the relative revenue of selfish miners is significantly reduced. Additionally, the proposed defense outperforms the tie-breaking defense, indicating its potential to strengthen the proof-of-work consensus algorithm.

Furthermore, the lower bound threshold metric is examined. In Figure 12, this metric is defined as the intersection point of the defense plots (*Tie-breaking*, *Nik Defense (VASLA)*, and *Nik Defense (ADTA)*) with the *Ideal Defense* plot. Evidently, the proposed defense using the ADTA increases the threshold from approximately 0.25 in tie-breaking to 0.4. The ADTA achieves this by effectively detecting when a decision is needed for a fork based on the weight or height parameter, enabling it to make informed decisions in unknown environments like blockchain.

