# OpenReview forum: "Adaptive Depth Tsetlin Automaton"
_ICLR.cc/2025/Conference — ICLR 2025 Conference Withdrawn Submission_

### Official Review · Reviewer_faz5 · 2024-11-03

**Soundness:** 2
**Presentation:** 2
**Contribution:** 2
**Rating:** 3
**Confidence:** 3

**Summary:**

The paper proposes an adaptive approach Tsetlin automation, a classic control problem inspired by the RL literature. The paper proposes an improvements on top of a combination of TA with RL, has showed theoretical improvements and guarantees and showcased improvements in toy settings against a few baseline methods.

**Strengths:**

The paper studies Tsetlin automation, a concept that's probably less adopted in the RL community overall. However, this is an exploration effort that can potentially be insightful to the RL community nevertheless.

**Weaknesses:**

It is not clear if in its current stage, the concept of combining Tsetlin automation with RL or more sophisticated deep RL system as proposed in this work is technically solid enough to be of interest to the conference community.

**Questions:**

### === Tsetlin automation as a framework in relation to RL ===

I am new to the concept to Tsetlin automation so would appreciate further elaboration from the authors. From reading the paper, I get the udnerstanding that Tsetlin automation is a formulation that seems to adapt a rather model-based approach to the decision making problem - i.e. assuming a MDP and applying a particular set of rule for action choosing and policy adaptation. This is in contrast to how RL formulation typically tries to allow the agent to self-adapt and learn to control in the environment. This hence leads to the combination of the two paradigms suggested by this work and some prior work, is that right?

Meanwhile, Tsetlin automation seems to focus on the agent's internal state rather than the external state possessed by the environment. So all the operations are performed on the inner state of the agent, and hopefully achieving a corresponding effect externally. This is in contrast to how modern deep RL approaches internal states, which are typically learned automatically. Does this sound like a fair characterization?

### === Experiments ===

In Fig 4, we see that the model performance improves linearly over time, and achieves gains over a few related baselines. How about comparing this TA integrated agent with regular RL agent that applies policy gradient or Q-learning. Since the latter are model free and TA assumes more structure with the agent, we should expect some improvements from the TA's side? I think a comparison will be useful for the readers from the RL community.

Do you see that bottleneck in adopting the TA formulation with general deep RL systems in more challenging high-d environments, such as with a learned internal states? The current experiments are quite limited to small toy domains and technically not solid enough in comparison to established model free or even model based deep RL algorithms.

What's the agent's activated time (plot (b)) in relation to the number of iteration the model takes in the environment? In general it might also be better if more texts are added to the figure captions to explain the xy axis.

---

### Official Review · Reviewer_swZT · 2024-11-04

**Soundness:** 2
**Presentation:** 1
**Contribution:** 2
**Rating:** 5
**Confidence:** 2

**Summary:**

This paper considers the Tsetlin Automaton (TA) setting, a single-state reinforcement learning model with a fixed depth parameter $N$. The authors propose the Adaptive Depth Tsetlin Automaton (ADTA) to dynamically modify the depth parameter $N$. They provide a theoretical analysis of ADTA and demonstrate that it can achieve better performance than the TA and two other baselines in a toy example. The authors also claim that ADTA can be applied to real-world scenarios.

**Strengths:**

- The paper discusses the problem of the fixed depth parameter $N$ in the TA setting, which is easy to understand.
- The authors propose a novel method, ADTA, with theoretical analysis.

**Weaknesses:**

- Overall, the writing has a lot of room for improvement. The structure of the paper is messy, and many important details are missed from the main page, making it hard to understand.
- The introduction includes problem definition and related work, which is not common practice and makes the introduction lengthy and unclear.
- The motivation for the TA setting is not clear, I would rather consider ADTA as a temporally-extended exploration method [1] in the bandit setting without introducing anything about the TA setting. I do not see the necessity of introducing the TA setting in the paper.
- The analysis of the ADTA lacks an integral structure. The authors define many notations and definitions, but since the proofs are in the appendix, it looks redundant and feels like the authors are trying to make the paper look more theoretical without providing a clear and integral analysis. I suggest the authors provide a more comprehensive and high-level analysis in the main page and put these definitions and notations into the appendix.
- The experiments are not convincing. Most experiments are put in the appendix, and only the results of a toy example are shown in the main page, without any introduction of the environment setting and parameter details, making it hard to understand the results.


[1] Dabney, Will, Georg Ostrovski, and André Barreto. "Temporally-extended {\epsilon}-greedy exploration." arXiv preprint arXiv:2006.01782 (2020).

**Questions:**

- What is the motivation of introducing the TA setting in the paper? What is the difference between the ADTA and the temporally-extended exploration method or other exploration methods in the bandit setting?
- The authors mention the practicality of the ADTA in real-world scenarios, but I am still confused about how the TA setting, with its assumption of single state and binary reward, can represent real-world scenarios. Can the authors provide more details about the real-world scenarios where ADTA can be applied?
- ADTA adapts a RL agent to dynamically adjust the depth parameter $N$. Why not directly use an RL agent to learn an optimal policy in the environment? What is the advantage of using ADTA with a dual-environment framework compared to directly learning an RL policy in the environment?
- Why not compare ADTA to other multi-armed bandit methods without the TA setting, such as UCB or Thompson Sampling? What is the advantage of ADTA compared to these methods, both experimentally and theoretically?
- Is the binary reward assumption a necessary condition for TA/ADTA?
- See Weaknesses.

---

### Official Review · Reviewer_4xCe · 2024-11-04

**Soundness:** 4
**Presentation:** 2
**Contribution:** 2
**Rating:** 3
**Confidence:** 4

**Summary:**

This paper proposes using an RL algorithm to adaptively change the depth of a Tsetlin Automation throughout training. The RL algorithm makes a choice to either making it deeper or shallower each time the action recommended by the TA changes. This approach is named ADTA. They present a proof of convergence of ADTA and explore its empirical performance.

**Strengths:**

This paper focuses on an approach to RL which has been relatively understudied in recent years. The clear formal model it provides, in comparison to Deep RL, offers a refreshing crispness and stability to the empirical and theoretical results. The approach is also relatively simple an natural.

**Weaknesses:**

There are three main weaknesses of this paper:
1) It is quite difficult to read, some key details are missing or unclear
2) The assumptions of the theory are quite strong and appear to me that they practically assume the conclusion
3) The experiments are not well explained in the main text and cannot be appropriately evaluated on the information provided.

I explain each of these weaknesses more in-depth below. Each of these weakness are of sufficent concern for me to reccomend rejection.

## Difficulty to Read

This paper is written with an Assumption that readers know the Tsetlin Automation which is likely false. Worse, the references for the Definition of Tsetlin Automation do not help. The first is in the original Russian and the second is a text book I could not find a copy of. Moreover, the idea is referenced to Sutton and Barto, though as far as I can tell it is not defined formally anywhere in Sutton and Barto, only mentioned in passing -- this seems to be a poor source to cite. I was able to infer the meaning from the text, but then I could not understand the first 2 pages until after reading section 1.2. Given that this will be the background of many readers, it is essential that the first 2 pages are written in a way to not assume that background.

Inaccuracies and inconsistencies in the exposition cause additional confusion throughout. There are a few points where the formalism switches back and forth between rewards and penalties. In Figure 1 the process seems to be maximising reward, but in Definition 1 it seems to be minimising penalties. This creates extra overhead every time I read a Definition as I need to infer if the rewards should be maximised or minimised based on context clues.

The type of transition labeled (iv) seems to be mis-defined, one of the definitions is a special case of the type of transition labeled (i). Since these were meant to be a partition, it seems that at least one of these is miss defined. Related, it is not clear why transition (2) of equation (3) is described as (iv) and not (i). It is also not clear what "indicates depth traversal" means on line 200. It's not clear what "Depth Transition" means in equation 4. This could be a result of the confusion about the types of transitions, but it is impossible for me to tell

In Section 2, the markers such as (i) and (1) are very overloaded in the first two paragraphs of Section 2. Which makes an already technically confusing section also linguistically confusing. I think I resolved what each was referring to, but it is possible I misunderstood because I guessed the wrong (i) being referenced.

An important ambiguity I have is that it is not clear when the high-level RL process takes over, or what it's observation space is. This seems central to the Definition, so it should really be the first thing that is described in Section 2. As a result it is not clear how to interpret the reward in equation 4, even past the uncertainty about what "Depth Transition" means.

The assumptions that underly the theorems, for instance having 2 actions, and Assumptions 1 and 2, should be described in the introduction to avoid giving a false impression of the generality of the results. The introduction set up expectations of theoretical depth to the results that were largely deferred to assumptions.


In Definition 5 "the RL agent interacts with the TA by switching from a current action to a new action". Is not sufficiently detailed, in that it doesn't say when the agent can choose and how that choice is constrained.

 In Assumption 1 "Negligible changes" is not the correct description of equation (12) it should say "no changes", which is a much stronger assumption. Assumption 1 should be mentioned in the statement of Lemma 1


On my first read, I thought that in Theorem 1 "will be able to learn" should be made formal, otherwise the theorem statement cannot be understood. However, after skimming through the appendix I realise this was supposed to be a reference to definition 4, but this was not clear as it used different words than definition 4. These words should made consistent and "will be able to learn" seems to be overstating the claim in Definition 4, since definition 4 only says that it will be able to do slightly better than an random initialisation which few would think of as really learning the task.


A few minor points that were also confusing:
	- The abstract calls TA a "single state RL model", when it has multiple states.
	- "designed to capture the intricate nature of human decision making and calculation" seems like quite an overstatement of TA.
	- on line 35, TA is described as having a "grid like configuration of n-actions and N parameters", when this is a grid of states not actions
	- Figure 1 should have a better description given the lack of background of most readers. Should describe the configuration, transitions, and assumed actions.
	- In definition 4 "capable to learn" is not grammatical.
	- Definition 3 is an remark or observation not a definition
	- Psi is first used in equation in equation 13 is not defined


## Extremely Strong Theoretical Assumptions

There is an expectation set in the introduction that we would theoretically understand something about the convergence behavior of ADTA above and beyond TA. However, assumption 1 apparently assumes that the part that ADTA introduces over TA converges. I read this assumption as saying that this ADTA eventually reduces to TA. Which means that, we essentially assume any results about TA eventually apply. I see this as assuming most of the result. The Theorem can then basically be seen as a theorem about TAs.

Still it would seem to be unclear what behavior a TA would converge to in general, thought the stable-state theorems of Markov chains imply they would usually converge to some stationary distribution. Still, Assumption 2 simplifies this further by essentially assuming that the internal environment has an absorbing state, which the Markov chain would go to. Together, Assumption 1 and 2 directly assume away the two difficult parts of the argument.


## The Experiments Cannot be Evaluated
The experiments as described in section 3 are not described in sufficient detail without reading the appendix, and reviewing the literature referenced there. Since the descriptions of these environments are completely deferred to the appendix, all that is known from the main text is that there exist 3 environments in which the method appears to outperform. As I do not understand the environments I cannot tell how significant reward or regret numbers are, or how overfit the algorithm may be to the setting. Moreover, to understand the experiments requires understanding the VASLA algorithm, which is also not described in the main text.

**Questions:**

Which empirical result in the appendix do you believe is most convincing of the empirical performance of ADTA?

---

### Official Review · Reviewer_FL8v · 2024-11-04

**Soundness:** 1
**Presentation:** 4
**Contribution:** 2
**Rating:** 3
**Confidence:** 3

**Summary:**

The paper tackles the problem of dynamically learning the depth parameter within the Tsetlin Automaton (TA) framework. The depth parameter plays an important role for calibrating exploration and exploitation of actions taken by the TA method.  Unlike previous approaches that fix the depth parameter, this framework allows the parameter to be learned adaptively using a reinforcement learning process while the TA component learns to transition between nodes based on the environment feedback.

**Strengths:**

The paper identifies an interesting gap in the current methodology and the need to address this issue is well motivated. Although the solution is very intuitive, the approach is complemented by both some theoretical analysis in addition to empirical analysis of the proposed method. The TA framework is also well-explained with both diagrams and text which make the workings of the framework clear.

The theoretical analysis is geared towards addressing a pressing question which is the convergence of the method given the introduction of an additional adaptive learner in the form of the RL component.

**Weaknesses:**

Although the paper seeks to deliver confidence in the convergence of the method, this does not seem rigorously developed and some questions remain about the validity of the authors' claims (see point 1).

Additionally, convergence aside, the theoretical analysis does not offer any insights about how the framework could improve performance and in which scenarios we can expect the framework to be of greatest benefit.


1, A concern that I have is that introducing an adaptive learner that affects the environment can make the environment appear non-stationary for the TA algorithm and therefore cause it not to converge. In turn, this can induce convergence issues for RL methods since such methods require stationarity of the environment. It looks to me that the role of Assumption 1 is to eradicate this issue, if that is indeed the case however, then the question of how and when Assumption 1 could be satisfied arises. The analysis would benefit from a discussion on how such a condition would be satisfied and what the role of the assumption in the analysis is. Additionally, it should be noted that proving that two learning processes converge individually is not sufficient for deducing that the two processes jointly converge, this observation occurs in for example reinforcement learning methods when applied to 2-player general sum games. Various analyses have been performed to try to understand this issue, for example, two-timescales analysis applied to actor-critic methods.

2. The paper does a nice job of listing various baselines from other related works. Although HCA is considered state of the art, it would still nonetheless be useful to see an empirical comparison against other leading  methods which is currently absent.

**Questions:**

1. In lines 200-202 it  is not  clear what transitions are being referred  to in the diagram.

2. In line 334 the object $C_{\rm ext}$ is introduced - what exactly is it?

3. In the definition of $c_{int -j}$ in equation 11, the number of transitions appears as an input. This seems to require knowledge of $T$ in advance - is this compatible with the general problem setting?

4. How does the framework perform with different RL methods?

---

### Note · Authors · 2024-12-12

I have read and agree with the venue's withdrawal policy on behalf of myself and my co-authors.